# ADAPTING IN THE DARK: TOWARDS STABLE AND EFFICIENT BLACK-BOX TEST-TIME ADAPTATION

## ABSTRACT

Test-Time Adaptation (TTA) for black-box models accessible only via APIs presents a significant yet largely unexplored challenge. Existing truly black-box methods are scarce; post-hoc output refinement shows minimal benefit, while naively introducing Zeroth-Order Optimization (ZOO) for prompt tuning at test time suffers from prohibitive query costs and catastrophic instability. To address these challenges, we introduce **BETA** (Black-box Efficient Test-time Adaptation), a novel framework that enables stable and efficient adaptation for both standard Vision Models and large Vision-Language Models. BETA uniquely employs a lightweight, local white-box *steering* model to create a tractable gradient pathway for optimization, circumventing the need for expensive ZOO methods. This is achieved through a prediction harmonization technique that creates a shared objective, stabilized by consistency regularization and a prompt learning-oriented filtering strategy. Requiring only *a single API call per test sample*, BETA achieves a +7.1% gain on a ViT-B/16 model and a +3.4% gain on powerful CLIP models; remarkably, its performance *surpasses* that of certain white-box and gray-box TTA methods (e.g., TENT and TPT). This practical effectiveness is further validated on a real-world commercial API, where BETA achieves a +5.2% gain for just $0.4—a 250x cost advantage over ZOO—establishing it as a robust and efficient solution for adapting models in the dark at test time. *Code will be released.*

## 1 INTRODUCTION

Modern deep learning models often face performance degradation when deployed in the wild due to distribution shifts between their training data and the target domain (Recht et al., 2019; Hendrycks & Dietterich, 2019b; Koh et al., 2021). Test-Time Adaptation (TTA) (Sun et al., 2020; Wang et al., 2021; Niu et al., 2023; Wang et al., 2022; Manli et al., 2022) has emerged as a crucial approach to address this challenge, aiming to adapt a pre-trained source model on-the-fly using unlabeled data from the target domain. While model providers typically handle general updates, TTA empowers users to develop stronger inference capabilities for fixed, pre-deployed APIs directly on their side, ensuring performance on specific user-defined data streams. The feasibility of TTA strategies, however, is determined by the level of access to the model. While white-box access allows full parameter and gradient manipulation (Wang et al., 2021; Niu et al., 2023), many state-of-the-art models are increasingly deployed as opaque, black-box APIs (Hurst et al., 2024; Achiam et al., 2023; Team et al., 2023). In this practical and restrictive setting, users can only provide an input and receive an output prediction, with no access to the model's architecture, parameters, or internal gradients (Sun et al., 2024; Tsai et al., 2020; Ouali et al., 2023).

TTA in this strictest black-box setting remains a largely unexplored and formidable challenge. Unlike offline transfer learning methods that rely on labeled support sets (few-shot) (Oh et al., 2023; Park et al., 2025), we focus on the strictly *online, source-free* setting where the model must adapt continuously to an unlabeled test stream. Recently, several backpropagation-free TTA methods have been proposed to eliminate the need for gradient propagation (Niu et al., 2024; Karmanov et al., 2024; Lee et al., 2025; Zhou et al., 2025). However, these approaches primarily target computational efficiency—such as reducing GPU memory usage—rather than addressing privacy or commercial constraints in black-box API scenarios (Niu et al., 2024; Meng et al., 2025). Consequently, these methods fall into a "gray-box" category, as they require access to internal model tokens or intermediate features (detailed comparison in Table 1). Truly black-box TTA methods applicable to both VMs

Table 1: Comparison of TTA methods across key capabilities. We evaluate each method's requirements for accessing model parameters, internal tokens, intermediate features, and gradients, alongside its visual encoder architectural flexibility, support for different model types (Vision models (VMs)/Vision-Language models (VLMs)), and query efficiency (One API call per test sample).

| Access | Method | w/o Params. | w/o Tokens | w/o Feats. | w/o Grad. | Arch-Agnostic | VMs | VLMs | 1 API/Sample |
|---|---|---|---|---|---|---|---|---|---|
| ☐ | TENT (Wang et al., 2021) | ✗ | ✗ | ✗ | ✗ | ✓ | ✓ | ✓ | ✓ |
| | TPT (Manli et al., 2022) | ✗ | ✗ | ✗ | ✗ | ✓ | ✗ | ✓ | ✓ |
| ▬ | T3A (Iwasawa & Matsuo, 2021) | ✗ | ✓ | ✗ | ✓ | ✓ | ✓ | ✓ | ✗ |
| | FOA (Niu et al., 2024) | ✓ | ✗ | ✗ | ✓ | ViT-only | ✓ | ✓ | ✗ |
| | B²TPT (Meng et al., 2025) | ✓ | ✗ | ✓ | ✓ | ViT-only | ✗ | ✓ | ✗ |
| | BCA (Zhou et al., 2025) | ✓ | ✓ | ✗ | ✗ | ✓ | ✓ | ✓ | ✓ |
| ■ | LAME (Boudiaf et al., 2022) | ✓ | ✓ | ✓ | ✓ | ✓ | ✓ | ✓ | ✓ |
| | Augmentation (Farina et al., 2024) | ✓ | ✓ | ✓ | ✓ | ✓ | ✓ | ✓ | ✗ |
| | Purification (Gao et al., 2023) | ✓ | ✓ | ✓ | ✓ | ✓ | ✓ | ✓ | ✗ |
| | ZOO | ✓ | ✓ | ✓ | ✓ | ✓ | ✓ | ✓ | ✗ |
| | **BETA (Ours)** | ✓ | ✓ | ✓ | ✓ | ✓ | ✓ | ✓ | ✓ |

and VLMs are scarce, as adaptation is constrained to only the model's inputs and outputs. While not originally proposed for this setting, methods like LAME are applicable because they often operate directly on output probabilities (Boudiaf et al., 2022). However, this post-hoc approach has limited adaptive capacity and often fails to provide consistent improvements, leaving the problem of robust black-box TTA largely open.

To address this critical gap, we explore the more powerful technique of learning an additive visual prompt in the input space (Bahng et al., 2022). The most straightforward solution is to employ Zeroth-Order Optimization (ZOO) (Liu et al., 2018; Spall, 1992; 1997; Hansen & Ostermeier, 2001; Hansen et al., 2003), a strategy we investigate as a baseline. However, we find this approach suffers from two critical limitations: prohibitively high query costs and catastrophic instability (Zhang et al., 2024b; Wang et al., 2024a). This instability arises because the optimization is driven by *noisy unsupervised signals* (e.g., entropy minimization) without true gradients. In high-dimensional input spaces, this creates a variance-heavy estimation that can lead to degenerate solutions, corrupting the model's representations rather than adapting them. For example, accuracy on the Contrast corruption collapses from 32.6% to as low as 4.1% with ZOO (Table 2). This motivates our development of a new approach that is both highly efficient—ideally requiring only a **single API call per test sample**—and robust against this optimization collapse.

We therefore propose **BETA (Black-box Efficient Test-time Adaptation)**, a novel framework that enables stable and efficient adaptation by leveraging a local, white-box *steering* model. Crucially, this steering model acts as a *local, client-side* guide initialized from *public* checkpoints (e.g., ImageNet), ensuring strict adherence to the black-box setting. It operates independently of the server-side API, requiring zero access to the proprietary target model's internals or training data, thus preserving complete privacy and security. Our initial analysis revealed that naively transferring gradients from the steering model is ineffective, as the gradient similarity between different architectures is near zero (see Fig. 2). This finding motivates our alternative approach, which moves beyond direct gradient approximation.

BETA's core mechanism is a *prediction harmonization* technique that fuses the outputs of the steering and target models, creating a shared, tractable optimization problem that is solved via a practical asymmetric gradient pathway. However, even with an efficient gradient signal, our preliminary analysis shows that the process of learning a prompt from random initialization remains highly unstable, leading to performance collapse (see Fig. 3). Therefore, this core mechanism is supported by two essential stabilization techniques to make the framework robust. We introduce a *consistency regularization* loss to prevent destructive prompt updates and a novel *prompt learning-oriented data filtering* strategy that provides a stable learning signal, distinguishing it from prior filtering methods designed for pre-trained normalization parameter updates (Niu et al., 2022; 2023).

Our extensive experiments validate BETA's effectiveness across various model architectures and real-world scenarios. On standard VMs, BETA achieves an average accuracy of 62.6% on ImageNet-C with ViT-B/16, a **+7.1% gain** over the source model. This result not only surpasses all black-box baselines but remarkably outperforms strong *white-box* methods like TENT (Wang et al., 2021) and CoTTA (Wang et al., 2022), all while requiring only a single API call per test sample versus 16 for ZOO-based approaches. This effectiveness extends to powerful VLMs; when adapting a black-box CLIP model, BETA boosts its average accuracy to 63.4%. This surpasses a suite of specialized *white-box* and *gray-box* methods developed for VLMs (e.g., TPT (Manli et al., 2022),

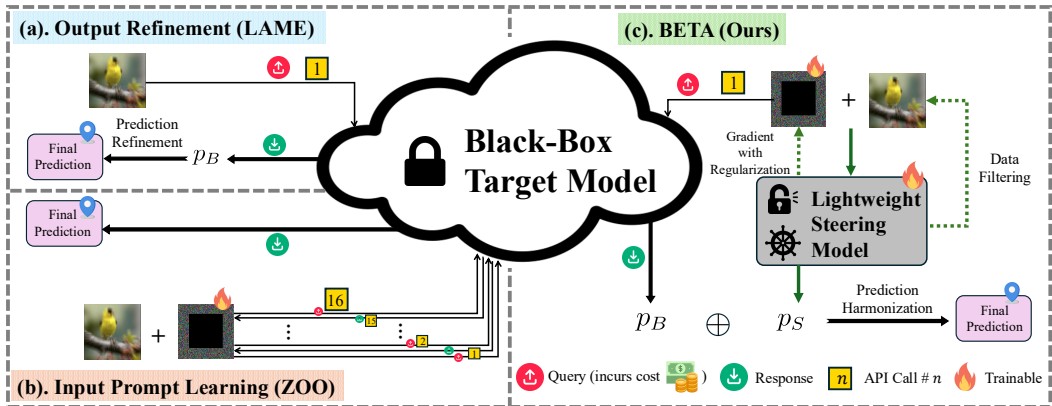

Figure 1: Comparison of black-box test-time adaptation strategies. **(a)** Output Refinement (LAME) is limited to post-processing predictions, while **(b)** ZOO-based Input Prompt Learning requires multiple expensive API calls for prompt optimization. In contrast, **(c)** BETA achieves efficient single-query adaptation by leveraging a lightweight steering model with prediction harmonization to create a tractable gradient pathway, stabilized through data filtering and regularization.

DynaPrompt (Xiao et al., 2025), and TCA (Wang et al., 2024b)), demonstrating BETA's unique capability in a domain previously unexplored in the strictest black-box setting. Finally, on a real-world commercial Clarifai API, BETA proves its immense practical value and cost-efficiency. It achieves a **+5.2%** performance gain with a budget of just **$0.4**, whereas a ZOO-based competitor requires over $100—a 250x greater cost—to reach a similar performance. At that same $100 budget, BETA's advantage widens significantly, delivering a substantial **+17.1% gain**.

**Main Findings and Contributions.** (1). We provide the first systematic evaluation of TTA in the strict, API-only Black-box setting. Our analysis confirms that existing applicable methods like post-hoc output refinement have limited adaptive capacity. We further establish input prompting with ZOO as a powerful but flawed baseline, revealing its critical inefficiency and optimization instability. (2). We introduce BETA, a novel framework that addresses challenges of inefficiency and instability in Black-box TTA. It bypasses expensive query-based optimization by using a lightweight steering model to enable an efficient gradient pathway via prediction harmonization, while consistency regularization and prompt-oriented data filtering ensure robust adaptation. (3). We establish a new state-of-the-art for black-box TTA. BETA not only significantly outperforms the ZOO-based baselines but also achieves performance competitive with and even *surpasses* strong white-box adaptation methods. Its practical effectiveness is validated on a real-world commercial API, where our *single-query-per-sample* approach demonstrates a 250x cost advantage over ZOO.

## 2 RELATED WORKS

**Test-time Adaptation (TTA).** TTA adapts pre-trained models on-the-fly with unlabeled target data to handle distribution shifts (Sun et al., 2020; Niu et al., 2023; 2022; Wang et al., 2022; Zhang et al., 2025a;b; Manli et al., 2022). Most works assume *white-box* access, enabling methods to directly update model parameters by minimizing prediction entropy or using consistency objectives (Wang et al., 2021; Niu et al.; 2023). Recent backpropagation-free methods have emerged for efficiency but typically operate in a *gray-box* setting, as they still require access to internal model representations like features or tokens, making them inapplicable to strict API-only scenarios (Niu et al., 2024; Meng et al., 2025; Zhou et al., 2025; Wang et al., 2024b; Lee et al., 2025). Truly *black-box* TTA remains a significant challenge, with applicable strategies limited to post-hoc output refinement that offers limited adaptive capacity (Boudiaf et al., 2022). In contrast, our work, BETA, addresses this gap by using a local steering model to enable efficient adaptation in the strict black-box setting, creating a tractable optimization pathway without requiring direct model access or expensive queries.

**Black-box Model Adaptation.** The adaptation of black-box models has been explored across various domains, including vision and language (Sun et al., 2024; Tsai et al., 2020; Oh et al., 2023; Liu et al., 2024; Sun et al., 2022), but typically for offline transfer learning with labeled data—a setting

with fundamentally different requirements from unsupervised, online TTA. A prominent approach in this area uses ZOO to learn input prompts that reprogram a model for a specific downstream task (Oh et al., 2023; Tsai et al., 2020; Liu et al., 2020). However, these ZOO-based methods are hindered by high query costs and optimization instability (Wang et al., 2024a; Oh et al., 2023). Other methods for VLMs often operate in a gray-box setting, requiring access to intermediate representations like text embeddings (Ouali et al., 2023; Wang et al., 2024a), which violates the strict black-box assumption. Beyond optimization-based methods, we also consider input-level heuristics. Test-Time Augmentation strategies are potential candidates, but existing methods often require prior training on labeled data (Shanmugam et al., 2021) or access to logits to adjust temperature (Farina et al., 2024), violating strict black-box constraints. While basic augmentation strategies can be adapted, they drastically increase API costs, scaling linearly with the number of augmentations (e.g., $64\times$ cost for standard protocols). Similarly, diffusion purification methods (Gao et al., 2023; Nie et al., 2022) utilize generative models to reconstruct inputs. While specific approaches like (Gao et al., 2023) require training a diffusion model on proprietary source data, employing an off-the-shelf diffusion model is a feasible workaround. However, the iterative nature of the reverse diffusion process results in high latency, making it unsuitable for fast, online adaptation. In contrast, our work is the first to tackle the unique challenges of *unsupervised, online* Test-Time Adaptation in this strict setting, where no labels are available and query efficiency is paramount.

## 3 METHOD

### 3.1 PROBLEM FORMULATION AND MOTIVATION

Test-Time Adaptation (TTA) aims to adapt a model $f$, pre-trained on a source domain, to an unlabeled target domain $\mathcal{D}_T = \{x_j^T\}_{j=1}^{|\mathcal{D}_T|}$ encountered during inference. In the common online setting, target data arrives as a stream of batches $\{B_t\}_{t=1}^T$, and the model is updated on-the-fly without ground-truth. The feasible adaptation strategies are determined by the level of access to the model $f$, which typically falls into one of three categories (Table 1):

- **White-Box Access** ( ☐ ): The full model architecture and all its parameters are accessible. This allows for the computation of gradients via backpropagation.
- **Gray-Box Access** ( ▨ ): Intermediate representations, e.g., internal tokens or features, are accessible, while the full computational graph and parameters remain hidden.
- **Black-Box Access** ( ■ ): The model is treated as an opaque API. The only possible interaction is to provide an input $x$ and receive a final output prediction $p(y|x) = f(x)$. No information about the model's architecture, parameters, or intermediate states is available.

**Existing Approaches and Their Limitations.** In the strict Black-Box TTA setting, existing methods primarily operate on either the model's output or its input space, each presenting distinct challenges for online API adaptation. Strategies that focus on **output refinement**, such as LAME (Boudiaf et al., 2022), are highly efficient as they operate post-hoc without requiring model queries. However, by working solely on the final predictions, their adaptive capacity is inherently limited, often resulting in marginal performance gains.

Conversely, methods that operate on the **input space** offer greater adaptive potential but frequently incur high costs or latency. Test-Time Augmentation (TTA) strategies (Shanmugam et al., 2021; Farina et al., 2024) enhance robustness by aggregating predictions across multiple augmented views; however, in an API setting, this linearly increases the query cost (e.g., $N$ views require $N$ paid API calls), reducing economic viability. Similarly, diffusion-based adaptation methods (Gao et al., 2023; Nie et al., 2022) effectively project inputs onto the source manifold but typically require iterative denoising steps, introducing significant latency that hinders real-time online applications. Finally, while Zeroth-Order Optimization (ZOO) (Niu et al., 2024) theoretically enables prompt learning without gradients, it is often hindered by high query complexity and optimization instability in the absence of ground-truth supervision.

### 3.2 BETA: BLACK-BOX EFFICIENT TEST-TIME ADAPTATION

These trade-offs motivate **BETA**, which seeks to combine the adaptive capacity of input prompting with the query efficiency of output-based methods. To address the inaccessibility of the target model's gradients while avoiding the high cost of ZOO, BETA operates using two distinct models:

- **Target Model ($f_B$):** The powerful, inaccessible black-box model (e.g., a remote API). We can only query it to get prediction $p_B(x)$.
- **Steering Model ($f_S$):** A lightweight, local white-box model (e.g., ViT-Small). We have full access to its parameters and gradients.

To adapt the black-box model without altering its weights, we learn an additive visual prompt $\delta \in \mathbb{R}^{H \times W \times C}$. This prompt is added to the input image $x$ to produce a prompted version $x' = x + \delta$. The goal is to optimize $\delta$ using gradients derived locally from $f_S$ to improve the predictions.

**The Challenge of Black-Box Prompt Optimization.** A powerful adaptation strategy is to learn an additive visual prompt, $\delta \in \mathbb{R}^{H \times W \times C}$, which is added to an input image $x$ to produce a prompted version $x' = x + \delta$. In a black-box setting, a straightforward approach to optimize this prompt is to employ ZOO to minimize the Shannon entropy of the model's predictions (Wang et al., 2021), $\mathcal{H}(p_B(x')) = -\sum_{c=1}^{C} p_B^c(x') \log p_B^c(x')$, where $p_B^c(x')$ is the model's predicted probability for class $c$. However, our investigation reveals two critical drawbacks: *prohibitively high query complexity* (e.g., a standard CMA-ES setup requires 28 API queries per test sample (Niu et al., 2024)) and *fundamental instability*. This instability stems from noisy unsupervised signals, e.g., entropy, which can cause the optimization to learn degenerate solutions that corrupt the input's semantic features to produce high-confidence but incorrect predictions. This leads to inconsistent performance and catastrophic collapse on challenging domains (e.g., on the Contrast corruption, accuracy collapses from 32.6% to 4.1%, 26.8%, and 12.7% across three ZOO methods in Table 2).

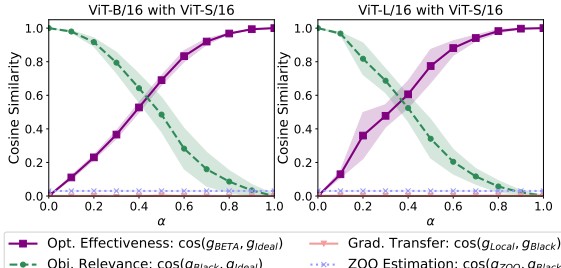

Figure 2: We analyze the trade-off between Objective Relevance (alignment with the true target gradient) and Optimization Effectiveness (alignment with the practical steering gradient) as a function of $\alpha$. The intersection of these opposing curves identifies the optimal range (e.g., $\alpha \in [0.4, 0.6]$) where the objective is simultaneously relevant to the target and tractable for optimization. The curves are plotted based on the validation sets of ImageNet-C.

### 3.3 PREDICTION HARMONIZATION

**From Naive Transfer to Harmonized Relaxation.** Our approach is motivated by the failure of direct estimation methods. To formalize our analysis, we introduce the notation $\nabla \mathcal{H}(p; \cdot)$ to denote the gradient of the entropy of a prediction $p$, computed by backpropagating through the specific model indicated by the second argument. Using this notation, our ultimate goal is to minimize the entropy of the black-box model, which implies following the **Black-box Model Gradient** $g_{\text{Black}} = \nabla \mathcal{H}(p_B; f_B)$. However, since $f_B$ is inaccessible, gradients cannot flow through it, rendering $g_{\text{Black}}$ intractable. Existing alternatives fail to provide a reliable substitute: ZOO suffers from prohibitive costs and instability, while naively transferring the **Local model Gradient** from a steering model ($g_{\text{Local}} = \nabla \mathcal{H}(p_S; f_S)$) is ineffective, as our analysis shows the gradient similarity between different architectures is consistently near zero ($\approx 0.0006$).

To overcome this, we relax the problem to finding a prompt that improves *both* models simultaneously. We define a **Harmonized Prediction**, $p_H$, that fuses the outputs of the steering model ($p_S$) and the black-box model ($p_B$) with a weighting parameter $\alpha \in [0, 1]$:

$$p_H(x') = \alpha \cdot p_S(x') + (1 - \alpha) \cdot p_B(x'). \tag{1}$$

Optimizing this shared objective presents a challenge. Theoretically, the ideal update direction, denoted as $g_{\text{Ideal}} = \nabla_\delta \mathcal{H}(p_H; f_S, f_B)$, requires backpropagating through the computational graphs of *both* the steering and target models. However, since the internal states of the black-box model $f_B$ are inaccessible, $g_{\text{Ideal}}$ is intractable. To address this, we employ an asymmetric optimization strategy: we approximate the ideal update by computing the gradient of the *same* harmonized objective but restricting the gradient flow exclusively to the steering model's pathway. This yields our tractable proxy, $g_{\text{BETA}} = \nabla_\delta \mathcal{H}(p_H; f_S)$, which allows us to target the joint harmonized distribution without requiring internal access to the black-box model.

**Empirical Justification.** To justify the use of $g_{\text{BETA}}$ as a valid proxy for the intractable $g_{\text{Ideal}}$, we conduct a comprehensive gradient analysis across four validation corruption domains. For this analysis only, we temporarily assume white-box access to the target black-box model to compute the otherwise inaccessible vectors ($g_{\text{Black}}$ and $g_{\text{Ideal}}$). Our analysis in Fig. 2 confirms that simpler strategies fail. The cosine similarity between the naive Local Gradient ($g_{\text{Local}}$) and the Target Gradient ($g_{\text{Black}}$) is consistently near zero. Similarly, ZOO gradient estimates are highly noisy in the one-step setting and prove no more effective than local transfer despite their high cost.

BETA's success is rooted in how the weighting parameter, $\alpha$, navigates a trade-off between two competing factors shown in Fig. 2. The first is **Objective Relevance**, which measures how well our tractable objective aligns with the true goal (Relevance$(\alpha) = \cos(g_{\text{Ideal}}, g_{\text{Black}})$). The second is **Optimization Effectiveness**, which measures how well our practical proxy can optimize this objective (Effectiveness$(\alpha) = \cos(g_{\text{BETA}}, g_{\text{Ideal}})$). These factors are in opposition: a low $\alpha$ yields high Relevance but negligible Effectiveness (as gradients cannot flow through $f_B$), while a high $\alpha$ yields perfect Effectiveness for an irrelevant objective. The success of BETA lies in identifying an optimal range for $\alpha$ (e.g., $[0.3, 0.5]$) where a principled compromise is struck. This confirms that BETA succeeds not by directly approximating the target gradient, but by constructing a shared optimization problem where the practical proxy $g_{\text{BETA}}$ effectively aligns with the ideal update direction $g_{\text{Ideal}}$.

### 3.4 STABILIZATION AND JOINT OPTIMIZATION

**Instability of Unconstrained Optimization.** While the harmonized objective provides a tractable gradient pathway, our investigation reveals that this process is inherently unstable when applied in isolation. To demonstrate this, we evaluated a baseline version using only the harmonized objective on the ImageNet-C Contrast domain. The results in Fig. 3 show that naively optimizing the randomly initialized prompt leads to either gradual decay or catastrophic collapse. This instability stems from noisy unsupervised signals, which can cause the optimization to learn degenerate solutions that corrupt the input's semantic features. To ensure robust adaptation, BETA incorporates two critical stabilization mechanisms.

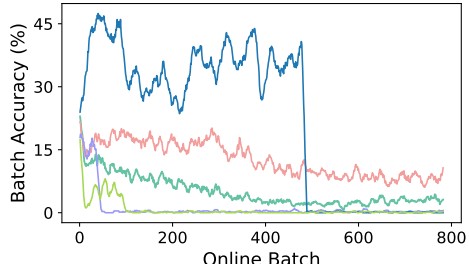

Figure 3: Five independent runs of using solely Eqn. (1), showing either performance collapse or failure to improve. Results obtained on ImageNet-C (Contrast, level 5).

**Prompt Learning-oriented Data Filtering.** The first step to ensuring stability is to filter the training signal. Our analysis indicates that updating the prompt using all incoming data degrades performance because high-entropy test samples provide noisy gradients. To ensure the prompt learns only from stable signals, we update it using samples with a prediction entropy $\mathcal{H}(p_S(x))$ below a threshold $\epsilon$. This filtering is integrated directly into the harmonization objective via a weight term $w_H(x')$:

$$\mathcal{L}_{\text{Harmon}}(x') = w_H(x')\mathcal{H}(p_H(x')), \tag{2}$$

where the weight filters out high-entropy samples and assigns a soft, confidence-based score to reliable ones: $w_H(x) = \frac{1}{\exp[\mathcal{H}(p_S(x))-\epsilon]} \cdot \mathbb{I}_{\{\mathcal{H}(p_S(x))<\epsilon\}}(x)$. Unlike methods that filter for pre-trained normalization parameters (Niu et al., 2022), we deliberately retain all reliable samples for the prompt update, as learning a visual prompt from a random initialization is a challenging optimization problem that benefits from more data.

**Consistency Regularization.** While filtering removes noisy samples, the optimization process itself requires regularization to prevent the catastrophic collapse observed in Fig. 3. Since prompts are randomly initialized, an unconstrained entropy objective can be minimized by learning degenerate solutions that destroy the model's representations. To prevent this, we introduce a consistency regularization that anchors the update to the model's reliable pre-trained knowledge by minimizing the KL-divergence between predictions on the clean ($x$) and prompted ($x'$) images:

$$\mathcal{L}_{\text{consist}}(x, x') := D_{\text{KL}}(p_S(x)\|p_S(x')) = \sum_{c=1}^{C} p_S^c(x) \log \frac{p_S^c(x)}{p_S^c(x')}. \tag{3}$$

Table 2: Classification accuracy (%) on ImageNet-C (severity 5) using **ViT-B/16** (87M) as the black-box model. BETA achieves the highest performance among black-box methods and outperforms several strong white-box approaches. *White-box and gray-box methods are shown for reference.*

| Access | Method | Noise | | | Blur | | | | Weather | | | | Digital | | | | Avg. | Gain |
|---|---|---|---|---|---|---|---|---|---|---|---|---|---|---|---|---|---|---|
| | | Gauss. | Shot | Impul. | Defoc. | Glass | Motion | Zoom | Snow | Frost | Fog | Bright. | Contr. | Elastic | Pixel. | JPEG | | |
| | Source | 56.8 | 56.8 | 57.5 | 46.9 | 35.6 | 53.1 | 44.8 | 62.2 | 62.5 | 65.7 | 77.7 | 32.6 | 46.0 | 67.0 | 67.6 | 55.5 | 0.0 |
| ☐ | TENT | 60.3 | 61.6 | 61.8 | 59.2 | 56.5 | 63.5 | 59.1 | 54.2 | 64.5 | 2.2 | 79.1 | 67.4 | 61.5 | 72.5 | 70.6 | 59.6 | +4.1 |
| | SAR | 59.1 | 60.5 | 60.6 | 57.1 | 55.6 | 61.5 | 57.4 | 65.8 | 63.4 | 67.4 | 78.7 | 62.6 | 62.2 | 72.0 | 70.2 | 63.6 | +8.1 |
| | CoTTA | 63.3 | 63.9 | 64.5 | 55.0 | 51.2 | 65.0 | 56.1 | 68.8 | 69.2 | 71.2 | 78.3 | 9.6 | 64.3 | 73.4 | 71.2 | 61.6 | +6.1 |
| | ETA | 60.9 | 62.2 | 62.2 | 59.5 | 57.4 | 63.6 | 60.1 | 68.3 | 65.8 | 71.5 | 79.3 | 66.9 | 64.9 | 72.9 | 71.1 | 65.8 | +10.3 |
| ▨ | T3A* | 56.4 | 56.9 | 57.3 | 47.9 | 37.8 | 54.3 | 46.9 | 63.6 | 60.8 | 68.5 | 78.1 | 38.3 | 50.0 | 67.6 | 69.1 | 56.9 | +1.4 |
| | FOA* | 57.0 | 58.5 | 57.8 | 51.7 | 35.0 | 37.1 | 27.2 | 20.2 | 11.9 | 72.2 | 76.8 | 0.6 | 39.1 | 66.7 | 67.0 | 44.9 | -10.6 |
| ■ | LAME | 56.5 | 56.5 | 57.2 | 46.4 | 34.7 | 52.7 | 44.2 | 58.4 | 61.5 | 63.1 | 77.4 | 24.7 | 44.6 | 66.6 | 67.2 | 54.1 | -1.4 |
| | ZOO-CMA | 58.2 | 59.6 | 60.3 | 50.8 | 38.6 | 55.2 | 45.7 | 58.5 | 59.6 | 59.7 | 76.7 | 4.1 | 49.8 | 71.2 | 70.0 | 54.5 | -1.0 |
| | ZOO-RGF | 59.6 | 58.7 | 60.4 | 47.7 | 37.8 | 53.5 | 44.6 | 58.2 | 61.7 | 63.4 | 76.7 | 26.8 | 49.4 | 70.7 | 70.2 | 56.0 | +0.5 |
| | ZOO-SPSA-GC | 59.6 | 58.7 | 60.2 | 47.9 | 38.0 | 53.7 | 44.7 | 58.2 | 61.7 | 63.6 | 76.7 | 12.7 | 49.4 | 70.7 | 70.2 | 55.1 | -0.4 |
| | TTA-Aug | 55.4 | 54.2 | 55.2 | 43.7 | 48.6 | 48.9 | 45.5 | 57.8 | 63.1 | 60.0 | 76.9 | 49.6 | 41.7 | 65.7 | 67.8 | 55.6 | +0.1 |
| | DDA | 64.7 | 65.0 | 64.6 | 46.3 | 41.3 | 51.7 | 43.7 | 59.1 | 61.3 | 45.0 | 74.9 | 40.6 | 54.4 | 72.2 | 68.4 | 56.9 | +1.4 |
| | **BETA (Ours)** | **60.5** | **60.7** | **61.1** | **54.5** | **52.2** | **59.9** | **56.3** | **63.6** | **64.7** | **66.1** | **78.1** | **53.4** | **62.1** | **73.3** | **72.0** | **62.6** | **+7.1** |

**Final Objective and Joint Optimization.** BETA operates in a strictly online, one-pass manner, performing a single gradient step for each incoming test batch $B_t$ to ensure minimal latency. The optimization targets are distinct: the Harmonization loss and Consistency regularization are minimized to update the visual prompt $\delta$, while the Steering loss exclusively updates the normalization parameters $\theta$ of the steering model. To maximize the steering model's effectiveness, we adapt these normalization layers using only samples that are both reliable and non-redundant (Niu et al., 2022). Non-redundancy is determined by comparing a sample's prediction against an exponential moving average of past predictions, $\bar{p}_{t-1}$, using a diversity margin $d$. The loss for the steering model's normalization parameters $\theta$ is defined as:

$$\mathcal{L}_{\text{Steer}}(x') = w_S(x')\mathcal{H}(p_S(x')), \tag{4}$$

where the weight $w_S(x)$ identifies the desired subset by filtering for samples that are both reliable (low entropy) and non-redundant (high cosine distance): $w_S(x) = \frac{1}{\exp[\mathcal{H}(p_S(x))-\epsilon]} \cdot \mathbb{I}_{\{\mathcal{H}(p_S(x))<\epsilon\}} \cdot \mathbb{I}_{\{|\cos(p_S(x),\bar{p}_{t-1})|<d\}}$. The effective number of samples contributing to the model update is determined dynamically per batch by these filtering weights ($w_H$ and $w_S$). The final objective for BETA combines the prompt-learning and the normalization-layer objectives, along with the consistency regularizer, averaged over the batch $B_t$:

$$\mathcal{L}_{\text{BETA}} = \mathbb{E}_{x \in B_t} \left[ \mathcal{L}_{\text{Harmon}}(x') + \mathcal{L}_{\text{Steer}}(x') + \lambda\mathcal{L}_{\text{consist}}(x,x') \right]. \tag{5}$$

# 4 EXPERIMENTS

**Datasets and Models.** We evaluate our method across several challenging benchmarks: ImageNet-C at severity level 5 (Hendrycks & Dietterich, 2019a), ImageNet-S (Sketch) (Wang et al., 2019), and ImageNet-R (Rendition) (Hendrycks et al., 2021). In our experiments, we treat powerful, large-scale models as the inaccessible black-box targets: standard Vision Transformers ViT-B/16 (87M parameters) and ViT-L/16 (304M), and the Vision-Language Model CLIP with a ViT-B/16 backbone (CLIP-B/16, 150M) (Dosovitskiy et al., 2021; Radford et al., 2021). Adaptation is guided by a much smaller, fully accessible ViT-S/16 (22M) steering model. To validate BETA in a practical, real-world scenario, we also test it using a commercial Clarifai[1] API, which charges $0.0032 per request.

**Compared Methods.** We conduct our comparison in the *source-free Fully TTA* setting, benchmarking against methods with varying levels of model access. For White-box methods, we include those applicable to both VMs and VLMs (Tent (Wang et al., 2021), T3A (Iwasawa & Matsuo, 2021), SAR (Niu et al., 2023), and CoTTA (Wang et al., 2022)), along with specialized approaches for VLMs (TPT (Manli et al., 2022), DynaPrompt (Xiao et al., 2025), and DPE (Zhang et al., 2024a)). For Gray-box methods, we compare against FOA (for both VMs and VLMs) (Niu et al., 2024) and others specific to VLMs (TDA (Karmanov et al., 2024), B²TPT (Meng et al., 2025), TCA (Wang et al., 2024b), BCA (Zhou et al., 2025), RA-TTA (Lee et al., 2025)). Our primary comparison is against truly **Black-box** methods: the post-hoc refinement method LAME (Boudiaf et al., 2022) and three ZOO baselines (CMA-ES, RGF, and SPSA) that we implemented to learn a visual prompt.

**Implementation Details.** For BETA, we set the weighting parameter $\alpha$ to 0.4. The shared visual prompt $\delta$ is trained with the AdamW optimizer using a learning rate of 0.01. We update only

---

[1] https://www.clarifai.com/

Table 3: Classification accuracy (%) on ImageNet-C (severity 5) using **ViT-L/16** (304M) as the black-box model. BETA achieves the best performance among black-box methods and outperforms several strong white-box approaches. *White-box and gray-box methods are shown for reference*.

| Access | Method | Noise | | | Blur | | | | Weather | | | | Digital | | | | Avg. | Gain |
|---|---|---|---|---|---|---|---|---|---|---|---|---|---|---|---|---|---|---|
| | | Gauss. | Shot | Impul. | Defoc. | Glass | Motion | Zoom | Snow | Frost | Fog | Bright. | Contr. | Elastic | Pixel. | JPEG | | |
| | Source | 62.5 | 62.0 | 63.3 | 52.9 | 45.3 | 60.7 | 55.2 | 66.0 | 62.3 | 62.6 | 79.9 | 40.1 | 56.2 | 74.3 | 72.8 | 61.1 | 0.0 |
| ☐ | TENT | 67.2 | 67.3 | 65.4 | 59.2 | 0.9 | 66.7 | 63.8 | 69.7 | 67.0 | 61.9 | 81.0 | 60.3 | 65.4 | 77.3 | 74.1 | 63.1 | +2.0 |
| | SAR | 65.6 | 66.7 | 66.9 | 58.6 | 57.8 | 60.5 | 61.0 | 69.3 | 67.0 | 68.1 | 81.0 | 60.2 | 61.8 | 76.8 | 74.3 | 66.4 | +5.3 |
| | CoTTA | 68.3 | 69.7 | 69.9 | 57.1 | 54.2 | 53.5 | 63.2 | 72.5 | 70.4 | 26.2 | 80.9 | 53.5 | 65.6 | 77.1 | 74.9 | 63.8 | +2.7 |
| | ETA | 67.4 | 58.3 | 67.9 | 63.4 | 61.3 | 67.7 | 62.9 | 70.7 | 68.4 | 66.3 | 81.3 | 54.0 | 66.0 | 77.7 | 74.1 | 67.2 | +6.1 |
| ▤ | T3A | 62.6 | 62.2 | 63.5 | 54.0 | 46.1 | 61.3 | 56.4 | 66.6 | 63.2 | 57.3 | 79.9 | 39.1 | 58.9 | 74.6 | 73.3 | 61.3 | +0.2 |
| | FOA[*] | 48.1 | 56.1 | 59.1 | 50.2 | 50.6 | 59.6 | 42.4 | 57.5 | 58.8 | 56.1 | 72.2 | 29.1 | 59.5 | 72.0 | 70.4 | 56.1 | -5.0 |
| ■ | LAME | 62.2 | 61.6 | 63.0 | 52.4 | 44.9 | 60.3 | 54.8 | 65.5 | 61.7 | 61.7 | 79.8 | 39.9 | 55.4 | 74.1 | 72.4 | 60.6 | -0.5 |
| | ZOO-CMA | 61.7 | 62.5 | 63.1 | 57.1 | 50.4 | 61.6 | 55.4 | 63.9 | 62.5 | 59.5 | 78.4 | 22.5 | 56.5 | 75.8 | 74.2 | 60.3 | -0.8 |
| | ZOO-RGF | 61.3 | 62.9 | 62.2 | 56.9 | 50.9 | 59.5 | 52.5 | 59.0 | 58.9 | 56.9 | 75.7 | 31.2 | 57.1 | 74.7 | 72.4 | 59.5 | -1.6 |
| | ZOO-SPSA-GC | 62.8 | 63.5 | 63.4 | 57.0 | 52.2 | 59.8 | 55.9 | 59.0 | 59.7 | 61.7 | 75.5 | 43.0 | 59.9 | 75.1 | 72.4 | 61.4 | +0.3 |
| | DDA | 68.0 | 68.3 | 68.0 | 52.8 | 49.8 | 59.3 | 53.8 | 64.3 | 63.4 | 55.8 | 78.0 | 46.9 | 61.1 | 76.4 | 73.1 | 62.6 | +1.5 |
| | **BETA (Ours)** | 63.1 | 64.0 | 63.5 | 59.7 | 55.1 | 63.6 | 59.4 | 66.1 | 65.0 | 66.2 | 80.0 | 55.1 | 65.0 | 76.2 | 74.5 | 65.1 | +4.0 |

the normalization layers of the local steering model using SGD with a learning rate of $2 \times 10^{-5}$. The weight for the KL consistency regularization $\lambda$ is set to 50, and we set the entropy threshold $\epsilon = 0.9 \times \ln(1000)$ for sample filtering. The visual prompt is structured as a padded frame with a width of 16 pixels, amounting to 39,936 learnable parameters, and is initialized from a Gaussian distribution. Additional experimental details are provided in Appendix A.

## 4.1 EXPERIMENTAL RESULTS

**Results on ImageNet-C with Vision Models.** Our main experiments evaluate BETA against a comprehensive suite of TTA methods on the ImageNet-C benchmark. We first test using a ViT-B/16 black-box model, with results for all white, gray, and black-box methods presented in Table 2 for a comprehensive comparison. The analysis reveals significant limitations in existing limited-access baselines. Gray-box methods like FOA[*] are inapplicable in our strict source-free setting, as their original design requires source statistics (Niu et al., 2024). In the black-box setting, LAME fails to improve upon the source model's performance. While ZOO-based methods can provide some benefit, they are inconsistent, collapsing on certain domains, and are highly inefficient, requiring 16 API calls per test sample versus BETA's single call. In stark contrast, BETA not only consistently improves performance across all domains but achieves an average accuracy of 62.6% (+7.1% gain). Remarkably, this surpasses all black-box baselines by a large margin and even outperforms several strong white-box methods such as TENT and CoTTA, approaching the accuracy of top performers like SAR, despite operating under much stricter access constraints.

This trend of superior performance continues when using the more powerful ViT-L/16, as shown in Table 3. Here, BETA again delivers the strongest performance among all black-box methods, achieving a +4.0% gain while ZOO-based approaches consistently degrade performance. This improvement is highly non-trivial and highlights the effectiveness of our steering mechanism. There is a substantial performance gap between the pre-trained steering model (ViT-S/16 at 39.5% accuracy) and the target black-box model (ViT-L/16 at 61.1% accuracy). Even when the steering model itself is fully adapted in a white-box setting, its performance is capped at 57.4% (detailed in Appendix Table 9). Yet, BETA successfully leverages this suboptimal steering model to guide the far more powerful ViT-L/16 to a new state-of-the-art black-box accuracy of 65.1%. This demonstrates that BETA is not simply relying on the local model's output, but is successfully discovering and transferring beneficial adaptation signals to the black-box model without requiring any internal access.

**Results on ImageNet-S and ImageNet-R.** To further evaluate BETA's generalization capabilities, we test its performance

Table 4: Results on ImageNet-S/R w.r.t. Acc (%).

| Access | Method | ViT-B/16 | | | CLIP (ViT-B/16) | | |
|---|---|---|---|---|---|---|---|
| | | Sketch | Rendition | Avg. | Sketch | Rendition | Avg. |
| | Source | 44.9 | 59.5 | 52.2 | 46.1 | 74.0 | 60.0 |
| ☐ | TENT | 49.1 | 63.9 | 56.5 | 49.5 | 75.3 | 62.4 |
| | SAR | 48.7 | 63.3 | 56.0 | 49.2 | 76.1 | 62.7 |
| | CoTTA | 50.0 | 63.5 | 56.8 | 50.4 | 75.6 | 63.0 |
| | TPT | – | – | – | 48.0 | 77.1 | 62.5 |
| | DynaPrompt | – | – | – | 48.2 | 78.2 | 63.2 |
| | DPE | – | – | – | 52.3 | 80.4 | 66.3 |
| ▤ | T3A | 48.5 | 58.0 | 53.3 | 49.1 | 75.6 | 62.4 |
| | FOA[*] | 44.7 | 59.2 | 52.0 | 45.8 | 73.2 | 59.5 |
| | TDA | – | – | – | 50.5 | 80.2 | 65.4 |
| | B²TPT | – | – | – | 49.5 | 78.6 | 64.1 |
| | RA-TTA | – | – | – | 50.8 | 79.7 | 65.3 |
| | TCA | – | – | – | 49.0 | 77.1 | 63.0 |
| | BCA | – | – | – | 50.9 | 80.7 | 65.8 |
| ■ | LAME | 44.4 | 59.0 | 51.7 | 45.4 | 72.8 | 59.1 |
| | ZOO-CMA | 44.7 | 58.8 | 51.8 | 45.6 | 72.5 | 59.1 |
| | ZOO-RGF | 44.4 | 58.1 | 51.3 | 45.3 | 72.1 | 58.7 |
| | ZOO-SPSA-GC | 45.1 | 59.3 | 52.2 | 46.0 | 72.8 | 59.4 |
| | **Ours** | 49.3 | 63.3 | 56.3 | 50.9 | 76.0 | 63.4 |

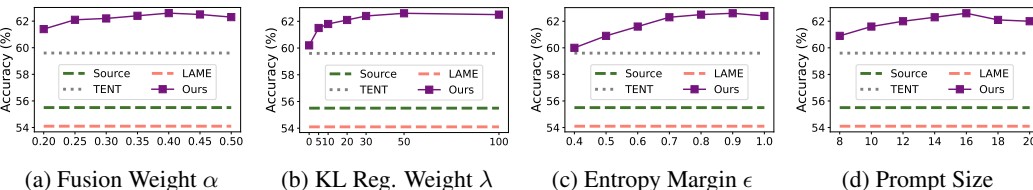

(a) Fusion Weight $\alpha$    (b) KL Reg. Weight $\lambda$    (c) Entropy Margin $\epsilon$    (d) Prompt Size

Figure 5: Sensitivity analysis of BETA's hyperparameters, showing stable performance across fusion weight $\alpha$ in Eq. 1, regularization weight $\lambda$ in Eq. 5, entropy margin $\epsilon$ in Eq. 2, and prompt size.

on ImageNet-S and ImageNet-R using ViT-B/16. The results in Table 4 demonstrate a consistent trend of strong performance. On both datasets, BETA significantly improves upon the source model's accuracy, achieving an average of 56.3%. This not only surpasses the black-box baselines but also outperforms strong white-box methods like T3A and SAR, underscoring our framework's robustness to diverse domain shifts. We then extend our evaluation to Vision-Language Models (VLMs), applying BETA to a CLIP model with a ViT-B/16 backbone.

To our knowledge, this is the first work to explore adaptation for powerful VLMs in the strictest, API-only black-box setting. The results in Table 4 highlight BETA's unique effectiveness in this challenging scenario. It is the only black-box method that can efficiently and effectively improve the pre-trained CLIP model, boosting its average accuracy to 63.4%. Remarkably, this black-box performance surpasses a suite of specialized white-box methods developed for VLMs, including TENT, SAR, TPT, and DynaPrompt, as well as gray-box methods such as TCA. This consistent success across different datasets and model types demonstrates that BETA is a general and powerful framework for black-box adaptation.

**Results on a Real-world API.** To validate BETA's practicality, we test it on a commercial Clarifai API, benchmarking performance against API cost in USD on the challenging ImageNet-C Contrast domain (Fig. 4). The results clearly show BETA's superior efficiency and effectiveness. With a budget of just $0.4—sufficient to adapt $\sim$ 120 test samples—BETA already improves upon the source model by +5.2%. In stark contrast, a query-intensive ZOO competitor requires over $100 to reach a similar performance, marking a **250x cost advantage** for our method. Furthermore, at that same $100 budget, BETA's advantage widens significantly, as it delivers a substantial +17.1% gain. This experiment demonstrates

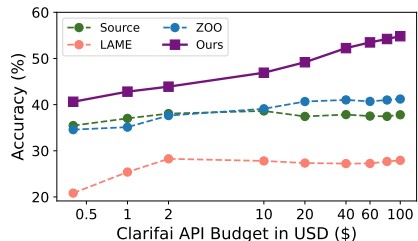

Figure 4: Performance vs. API budget on the Real-world Clarifai API.

BETA's significant real-world utility, making it a practical and effective solution for adapting commercial API-based models.

## 4.2 ABLATION STUDIES

**Hyperparameter Sensitivity.** We analyze BETA's sensitivity to its key hyperparameters in Fig. 5. Our analysis of the fusion weight $\alpha$ in Eq. 1 shows that the framework's performance is empirically robust, exhibiting stable and high performance across a wide range of values from 0.3 to 0.5 (Fig. 5a). The KL regularization weight $\lambda$ in Eq. 5 is shown to be a critical component; without it ($\lambda = 0$), performance is suboptimal as the prompt can learn degenerate solutions. As shown in Fig. 5b, performance improves significantly with the introduction of regularization and stabilizes across a broad range of $\lambda$ values from 20 to 100. For the entropy margin $\epsilon$ in Eq. 5, our results show that BETA performs robustly with a more lenient margin (tested from $0.4 \cdot \ln(1000)$ to $1.0 \cdot \ln(1000)$). Unlike methods adapting pre-trained parameters, learning a prompt from random initialization requires more data, making a less restrictive filter beneficial (Fig. 5c). Finally, for the prompt size (Fig. 5d), which corresponds to the frame width, we observe a clear trade-off: smaller prompts may lack the capacity to capture the domain shift, while larger prompts are harder to optimize. The performance peaks around a width of 16 pixels and remains stable across the tested range of 8 to 20.

Table 6: Effect of steering model choice. The `Source` and `TENT`-adapted accuracy of each local steering model are provided as a reference against the `BETA` accuracy on the large black-box models.

| Dataset | Black-Box Model | Source | LAME | ZOO | ViT-Tiny (6M) | | | ResNet50 (26M) | | | ViT-Small (22M) | | |
|---|---|---|---|---|---|---|---|---|---|---|---|---|---|
| | | | | | Source | TENT | BETA | Source | TENT | BETA | Source | TENT | BETA |
| ImageNet-C | ViT-B/16 (87M) | 55.5 | 54.1 | 56.0 | 21.4 | 22.0 | 58.2 | 24.2 | 31.4 | 60.8 | 39.5 | 51.9 | 62.6 |
| ImageNet-Sketch | ViT-B/16 (87M) | 44.9 | 44.4 | 45.1 | 20.9 | 21.3 | 45.2 | 27.9 | 29.7 | 47.5 | 32.8 | 35.6 | 49.3 |
| | CLIP-B/16 (150M) | 46.1 | 45.4 | 46.0 | | | 47.0 | | | 48.7 | | | 50.9 |
| Average | - | 48.8 | 48.0 | 49.0 | 21.1 | 21.7 | 50.1 | 26.7 | 30.2 | 52.3 | 35.0 | 41.0 | 54.3 |

**Analysis of BETA's Components.** We conduct an ablation study to dissect the contribution of each component in BETA, with results summarized in Table 5. Our analysis first reveals that strategies focusing solely on output adaptation (**Out-Adapt**) are insufficient. Both LAME's Prediction Refinement (**PR**) and our Prediction Harmonization (**PH**) strategy used in isolation (Exp-1) fail to improve upon the source model, demonstrating that effective black-box TTA requires input adaptation (**In-Adapt**). However, naively adding an input prompt (Exp-2) leads to a performance collapse to 51.6% accuracy. This highlights the inherent instability of learning a randomly initialized prompt without supervision—a task significantly more challenging than adapting well-initialized

Table 5: Component analysis on ImageNet-C.

| Method | In-Adapt | KL Reg. | Filt. | Out-Adapt | Acc. | Gain |
|---|---|---|---|---|---|---|
| Source | - | - | - | - | 55.5 | 0.0 |
| LAME | - | - | - | PR | 54.1 | -1.4 |
| ZOO | ✓ | - | - | - | 56.0 | +0.5 |
| Exp-1 | - | - | - | PH | 54.2 | -1.3 |
| Exp-2 | ✓ | - | - | PH | 51.6 | -3.9 |
| Exp-3 | ✓ | ✓ | - | PH | 59.7 | +4.3 |
| Exp-4 | ✓ | - | ✓ | PH | 60.2 | +4.7 |
| **BETA** | ✓ | ✓ | ✓ | PH | **62.6** | +7.1 |

normalization layers. Our stabilization techniques are designed to resolve this instability. Introducing either KL regularization (**KL Reg.**) in Exp-3 or sample filtering (**Filt.**) in Exp-4 provides a substantial performance boost, improving accuracy to 59.7% and 60.2%, respectively. The full BETA framework, which integrates both complementary techniques, achieves the best performance of 62.6%. This confirms that both stabilization mechanisms are essential for robust prompt learning.

**Effect of Steering Model Choice.** We investigate how the choice of the local steering model affects BETA's performance, with detailed results summarized in Table 6. Our analysis confirms that BETA is a flexible framework that consistently improves upon the source model across different steering models, including those with different sizes and architectures. Notably, even with a model as small as a 6M-parameter ViT-Tiny, our method successfully boosts the performance of both large black-box models (87M and 150M). Furthermore, the framework demonstrates strong cross-architecture generalization, as a CNN-based ResNet-50 can effectively improve the Transformer-based ViT and CLIP models. The improvement from BETA is highly non-trivial and goes far beyond the capabilities of the steering models themselves, a finding that holds true across all tested configurations. On average, our strongest steering model (ViT-Small), even when fully adapted with TENT, only reaches an accuracy of 41.0%—well below the 48.8% starting accuracy of the black-box models. Despite this, BETA successfully leverages these weaker models to support the black-box models to a final average accuracy of 54.3%. This demonstrates that BETA is not simply relying on the local model's output but is effectively discovering and transferring beneficial adaptation signals to successfully adapt large-scale models in the dark.

## 5 CONCLUSION

In this work, we addressed the critical challenge of adapting powerful models in the strict black-box setting where only API access is available. We introduced **BETA**, a novel framework that enables efficient and stable Test-Time Adaptation by leveraging a lightweight white-box steering model. The core of our method is a prediction harmonization technique that creates a tractable, shared objective, which is made robust through consistency regularization and a prompt-oriented data filtering strategy. Our extensive experiments show that BETA significantly outperforms existing black-box methods, achieves performance competitive with strong white-box approaches on both Vision and Vision-Language models, and demonstrates immense practical value on a commercial API with a 250x cost advantage over ZOO-based techniques. By demonstrating that a smaller, local model can effectively steer a powerful, inaccessible one, our work makes robust black-box TTA a practical reality and opens up new possibilities for adapting models in the dark at test time.

ETHICS STATEMENT

Our work adheres to the ICLR Code of Ethics. It relies on publicly available datasets and models and does not introduce any foreseeable societal risks.

REPRODUCIBILITY STATEMENT

To ensure reproducibility, we provide full implementation details in the main paper and appendix. We will release our source code publicly upon publication.

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

# **Appendix**

## A ADDITIONAL EXPERIMENTAL DETAILS

### A.1 BASELINES AND IMPLEMENTATION DETAILS

We compare BETA against a comprehensive suite of baselines with varying levels of model access, including white-box, gray-box, and black-box methods.

The following methods are applicable to both standard Vision Models (VMs) and Vision-Language Models (VLMs):

**Tent** (Wang et al., 2021) is a **white-box** method for fully test-time adaptation, which adapts a pre-trained model to a new test distribution without requiring any source data. The core idea is to encourage model confidence on the unlabeled test data by minimizing the Shannon entropy of its predictions for each incoming batch. To achieve this efficiently, Tent does not update the entire model; instead, it exclusively adapts the parameters within the model's normalization layers. For each test batch, it first updates the normalization statistics during the forward pass and then optimizes the learnable channel-wise affine transformation parameters via backpropagation on the entropy loss.

**SAR** (Niu et al., 2023) is a **white-box** method designed to stabilize online Test-Time Adaptation in challenging "wild" scenarios, such as with mixed domain shifts or small batch sizes, where standard entropy minimization can fail. The method identifies that model collapse during adaptation is often caused by noisy test samples producing large, disruptive gradients. To mitigate this, SAR employs a two-part strategy: it first filters out unreliable, high-entropy samples to reduce noise. For the remaining data, it then uses a sharpness-aware optimizer to guide the model parameters into a flat region of the loss landscape, enhancing robustness against any remaining noisy updates.

**Continual Test-Time Adaptation (CoTTA)** (Wang et al., 2022) is a **white-box** method designed to adapt models to continually changing target domains, addressing the challenges of error accumulation and catastrophic forgetting. To generate more reliable pseudo-labels, it employs a teacher-student framework where the student model is updated based on the weight-averaged and augmentation-averaged predictions of the teacher. To prevent catastrophic forgetting over long-term adaptation, CoTTA stochastically restores a small fraction of the student model's weights to their original source-trained values during the update process. The method is designed to adapt all parameters of the network.

**Test-Time Template Adjuster (T3A)** (Iwasawa & Matsuo, 2021) is a **gray-box** method for domain generalization that adapts a model's final linear classifier at test time. The method is backpropagation-free and works by first computing class-specific "pseudo-prototype" representations from the features of unlabeled test data. Once these prototypes are established, it classifies each new test sample based on its distance to these dynamically adjusted prototypes. This allows the model to leverage information from the target domain without requiring extensive optimization or altering the core feature extractor.

**Forward-Optimization Adaptation (FOA)**[2] (Niu et al., 2024) is a **gray-box** method designed for test-time adaptation in scenarios where backpropagation is infeasible, such as on quantized models or edge devices. The approach is entirely training-free and avoids modifying model weights by learning an additive input prompt using a derivative-free optimizer (CMA-ES). To guide this optimization, FOA introduces a novel fitness function that combines prediction entropy with a term measuring the statistical discrepancy between the test sample's activations and pre-computed source data activations. The framework also includes a "back-to-source" activation shifting scheme that directly modifies the final layer's features during the forward pass to better align them with the source domain.

**LAME** (Boudiaf et al., 2022) is a **black-box** method for online test-time adaptation that operates without requiring access to model parameters or gradients. Instead of adapting the network's weights, it adapts the model's output probabilities directly for a given batch of test data. The method

---

[2]FOA uses entropy minimization instead of activation discrepancy for source-free settings where source statistics are unavailable in our experiments.

proposes a Laplacian Adjusted Maximum-likelihood Estimation (LAME) objective, which finds the optimal latent class assignments by maximizing the data likelihood while being regularized by a Laplacian term that encourages label consistency among neighboring samples in the feature space. This objective is optimized efficiently using a concave-convex procedure and does not require back-propagation.

In contrast to the methods above, the following baselines are designed specifically for the adaptation of Vision-Language Models:

**Test-Time Prompt Tuning (TPT)** (Manli et al., 2022) is a **white-box** method that adapts Vision-Language Models like CLIP using only a single unlabeled test sample. For each test image, TPT creates multiple augmented views and optimizes a learnable text prompt via backpropagation to enforce prediction consistency across them. The optimization is guided by minimizing the entropy of the averaged predictions, and a confidence selection module filters out noisy augmentations that yield low-confidence outputs. TPT performs a one-step update on the prompt for each test sample.

**Dual Prototype Evolving (DPE)** (Zhang et al., 2024a) is a **white-box** method that performs test-time adaptation for VLMs by accumulating task-specific knowledge from both visual and textual modalities. The method maintains and evolves two sets of class prototypes—one textual and one visual—which are updated online as more test samples are processed. For each individual test sample, DPE learns temporary residual parameters to adjust both sets of prototypes. This sample-specific optimization is guided by a dual objective that encourages prediction consistency across augmented views and enforces alignment between the textual and visual prototypes for each class.

**DynaPrompt** (Xiao et al., 2025) is a **white-box** method that improves online test-time prompt tuning by leveraging information from previous test samples while mitigating the problem of prompt collapse. The core of the method is an online prompt buffer containing a set of learnable prompts that evolve over time. For each new test sample, DynaPrompt employs a dynamic selection strategy based on prediction entropy and probability difference to choose a relevant subset of prompts from the buffer for optimization. To adapt to new data, the framework also dynamically appends new prompts to the buffer and removes inactive ones.

**B$^2$TPT** (Meng et al., 2025) is a **gray-box** method that addresses test-time prompt tuning for black-box Vision-Language Models (VLMs) where gradients are inaccessible. To overcome this, it employs a derivative-free algorithm (CMA-ES) to optimize low-dimensional "intrinsic prompts," which are then projected into the full prompt space to make the high-dimensional optimization tractable. For supervision, the framework uses a "Consistent or Confident" (CoC) pseudo-labeling strategy to generate labels from the model's outputs. The method jointly optimizes text and vision prompts using a frozen CLIP ViT-B/16 backbone.

**Training-free Dynamic Adapter (TDA)** (Karmanov et al., 2024) is a **gray-box** method designed for efficient test-time adaptation of Vision-Language Models without requiring backpropagation. The method constructs a lightweight key-value cache during inference, which is progressively updated with incoming test samples. This cache consists of two components: a positive cache that stores image features and their corresponding high-confidence pseudo-labels, and a novel negative cache that stores negative pseudo-labels to improve robustness against label noise. The final prediction is a combination of the original CLIP output and the predictions derived from both the positive and negative caches.

**Retrieval-Augmented TTA (RA-TTA)** (Lee et al., 2025) is a **gray-box** method that adapts Vision-Language Models by incorporating external knowledge from a large image database at test time. Instead of a direct image-to-image search, RA-TTA uses a novel description-based retrieval process to find more relevant external images. For a given test image, it first identifies its most prominent visual features by selecting matching fine-grained text descriptions from a pre-compiled library. These selected text descriptions are then used as queries to retrieve semantically similar images from the database, and the VLM's initial prediction is refined using a relevance score derived from this external knowledge.

**Bayesian Class Adaptation (BCA)** (Zhou et al., 2025) is a **gray-box** method that adapts Vision-Language Models by updating both the class likelihood and prior at test time. It frames the adaptation problem using Bayes' theorem, identifying that existing methods only adapt the likelihood (class embeddings) while overlooking the class prior, which can shift in new domains. BCA em-

ploys a dual-update mechanism: it adapts the likelihood by updating the most relevant class embedding with an incoming visual feature via a running average. Concurrently, it adapts the prior by using the model's posterior prediction for the current sample to update the prior distribution of the predicted class, allowing the model to learn the new class frequencies on the fly.

**Token Condensation as Adaptation (TCA)** (Wang et al., 2024b) is a **gray-box** method that provides an efficient, training-free solution for test-time adaptation in Vision-Language Models. The method uniquely repurposes token condensation, a technique originally for improving ViT efficiency, as an adaptation mechanism. It introduces a domain-aware token reservoir that stores reliable class tokens from past test samples to serve as domain anchors. These anchors guide both a cross-head token condensation process, which prunes irrelevant visual tokens, and a logits self-correction mechanism that refines the model's final prediction.

## A.2 Detailed Analysis of Model Accessibility and Security Constraints

In this section, we provide a rigorous definition of the black-box setting adopted in this work. While prior literature often conflates different levels of restricted access, we draw sharp distinctions between access to *raw logits*, *softmax probabilities*, and *hard predictions*. This distinction is critical for evaluating the practical applicability of Test-Time Adaptation (TTA) methods on real-world commercial APIs.

**Mathematical Definitions of Output Levels.** Let $f_\theta(x)$ denote the pre-trained model. We distinguish between three specific levels of output granularity: 1). *Raw Logits (z):* The pre-activation output vector $z \in \mathbb{R}^C$, where values are unbounded ($-\infty < z_i < \infty$) and unnormalized. 2). *Softmax Probability Vector (p):* The normalized output distribution obtained via the softmax function $\sigma(\cdot)$, such that $p = \sigma(z) \in [0,1]^C$ with $\sum_i p_i = 1$. 3). *Top-1 Hard Prediction ($\hat{y}$):* A single scalar value representing the class index with the highest confidence, $\hat{y} = \arg\max_i p_i$, often accompanied by a single confidence score.

**Real-World API Protocols.** To determine the most realistic setting for black-box adaptation, we analyze standard commercial Machine Learning APIs (e.g., OpenAI (Hurst et al., 2024), Clarifai, Google Cloud Vision).

- *Why not Raw Logits?* Access to $z$ is frequently restricted as a security measure. Raw logits contain rich information regarding inter-class relationships ("dark knowledge") that significantly facilitates Model Extraction attacks and Knowledge Distillation (Hinton et al., 2015). By hiding $z$, API providers mitigate the risk of model theft.

- *Why Softmax Probabilities?* Most commercial APIs return the probability distribution $p$ rather than a single hard label $\hat{y}$. This is because downstream users typically require confidence estimates to make informed decisions (e.g., thresholding low-confidence predictions).

**Justification for BETA's Setting.** Based on these protocols, we define the strict *Black-Box* setting as one where the *Softmax Probability Vector* $p$ is available, but *Raw Logits* $z$ are hidden. This setting strikes the balance found in real-world deployments: it provides more information than the restrictive *Label-Only* setting (which only provides $\hat{y}$), enabling unsupervised objectives like entropy minimization ($H(p) = -\sum p_i \log p_i$). In contrast, we classify methods that require access to raw logits $z$ (e.g., for temperature scaling $z/\tau$ or re-normalization (Farina et al., 2024)) as *Gray-Box*. While these methods do not require gradients, they rely on information often hidden in secure deployment environments.

## B Additional Experimental Results

Table 7: Performance comparison on ImageNet Variants with CLIP-B/16. BETA outperforms strong augmentation-based and gray-box baselines while requiring only a single API call per image.

| Method | IN-Sketch | IN-R | IN-A | IN-v2 | ImageNet | Avg. Acc | Gain | # API/Img |
|---|---|---|---|---|---|---|---|---|
| Source | 46.1 | 74.0 | 47.9 | 60.9 | 66.7 | 59.1 | - | 1 |
| LAME | 45.4 | 72.8 | 48.1 | 61.6 | 66.7 | 58.9 | -0.2 | 1 |
| ZOO-SPSA-GC | 46.0 | 72.8 | 50.2 | 61.5 | 65.8 | 59.3 | +0.1 | 16 |
| $B^2$TPT (w/ tokens) | 49.5 | 78.6 | 55.3 | 65.4 | 69.6 | 63.7 | +4.6 | 120 |
| ZERO (w/ logits) | 48.4 | 77.2 | 59.6 | 64.2 | 69.3 | 63.7 | +4.6 | 64 |
| ZERO_ensemble (w/ logits) | 50.6 | 80.8 | 62.8 | 65.2 | 71.2 | 66.1 | +7.0 | 448 |
| **BETA (Ours)** | **50.9** | **76.0** | **62.8** | **65.1** | **77.5** | **66.5** | **+7.4** | **1** |

Table 8: Performance on the fine-grained EuroSAT dataset with CLIP-B/16. BETA achieves significant gains (+11.3%) with high efficiency.

| Method | Accuracy (%) | Gain (%) | # API/Img |
|---|---|---|---|
| Source | 42.0 | - | 1 |
| $B^2$TPT (w/ tokens) | 46.8 | +4.8 | 120 |
| ZERO (w/ logits) | 39.6 | -2.4 | 64 |
| ZERO_ensemble (w/ logits) | 43.8 | +1.8 | 448 |
| **BETA (Ours)** | **53.3** | **+11.3** | **1** |

## B.1 BETA'S PERFORMANCE ON OTHER IMAGENET VARIANTS AND EUROSAT

To provide a comprehensive evaluation, we extend our comparisons to include augmentation-based strategies and recent methods tailored for Vision-Language Models (VLMs). Specifically, we compare BETA against **ZERO** Farina et al. (2024), a test-time augmentation method that optimizes temperature using input augmentations. We note that while ZERO requires access to raw logits—violating strict black-box API constraints that typically only provide probabilities—we grant it this access for a rigorous upper-bound comparison. We evaluate both the standard ZERO (64 calls/image) and **ZERO_ensemble** (448 calls/image, using 7 text templates). We also include $B^2$**TPT** Meng et al. (2025), a recent prompt tuning method for VLMs.

**Classification of B$^2$TPT as Gray-Box.** We categorize B$^2$TPT as a gray-box method because it operates by modifying inputs in the embedding space. Specifically, it prepends learnable vectors directly to the text and image embeddings ($e_t$ and $e_v$), requiring internal access to the model's intermediate feature representations. This contrasts with the strict black-box setting of commercial APIs, which accept only raw image or text inputs. Furthermore, its underlying optimization (CMA-ES) is query-intensive, requiring approximately 120 API calls per input.

**Results on ImageNet Variants and EuroSAT.** We evaluate these baselines on the full suite of ImageNet variants (ImageNet-S, R, A, v2, and standard ImageNet) and the challenging fine-grained EuroSAT dataset. The results are summarized in Table 7 and Table 8.

BETA consistently outperforms these query-intensive baselines while maintaining strict API efficiency. On the ImageNet variants (Table 7), BETA achieves the highest average accuracy of 66.5%, surpassing the ensemble version of ZERO (66.1%) which requires 448 API calls per image. The efficiency gap is even more pronounced on EuroSAT (Table 8), where BETA achieves a substantial gain of +11.3% over the source model with a single API call, whereas augmentation baselines struggle or yield marginal gains despite their high computational cost. This demonstrates that BETA's effectiveness stems from learned adaptation rather than simple data augmentation, making it a far more practical solution for real-world deployment where API costs and rate limits are critical constraints.

Table 9: White-box TTA performance on the ViT-Small steering model on ImageNet-C. The results show that even when fully adapted, the steering model's performance is capped well below that of the unadapted black-box target models, highlighting the effectiveness of our steering mechanism.

|  | Source | TENT | T3A | SAR | CoTTA | LAME |
|---|---|---|---|---|---|---|
| **Avg.** | 39.5 | 51.9 | 40.4 | 57.4 | 46.0 | 38.9 |
| **Gain** | 0.0 | +12.4 | +0.9 | +17.9 | +6.5 | -0.6 |

Table 10: Comparison between Test-Time Knowledge Distillation (KD) and BETA on ImageNet-C. While KD is upper-bounded by the teacher's performance, BETA successfully adapts the black-box model to surpass its original baseline.

| Model Role | Architecture | Method | Avg. Acc (%) |
|---|---|---|---|
| Local Steering Model | ViT-S/16 | Source | 39.5 |
|  |  | TENT | 51.9 |
|  |  | KD (from ViT-B/16) | 50.3 |
| Black-Box Target Model | ViT-B/16 | Source | 55.5 |
|  |  | **BETA (Ours)** | **62.6** |

## B.2 LOCAL STEERING MODEL BASELINES

### B.2.1 WHITE-BOX TTA PERFORMANCE ON STEERING MODEL.

To demonstrate that BETA's improvement is non-trivial and not simply a result of relying on the steering model's outputs, we present the white-box adaptation performance of the ViT-Small steering model in Table 9. There exists a substantial performance gap between the pre-trained steering model (39.5% accuracy on ImageNet-C) and the target black-box models (e.g., ViT-L/16 at 61.1% accuracy). Even when the steering model itself is fully adapted in a white-box setting with a strong method like SAR, its performance is capped at 57.4%. This is still well below the starting accuracy of the black-box model it is meant to guide. This highlights that BETA successfully leverages this weaker, suboptimal steering model not for its direct predictions, but to discover and transfer beneficial adaptation signals to the far more powerful black-box model without requiring any internal access.

### B.2.2 COMPARISON WITH TEST-TIME KNOWLEDGE DISTILLATION

A natural question arises as to whether BETA's improvements stem from simply distilling the powerful black-box model's knowledge into the local steering model. To investigate this, and to verify that our framework is not merely performing Test-Time Knowledge Distillation (KD), we implemented a KD baseline following the protocol in (Zhao et al., 2024). Specifically, we employed the black-box ViT-B/16 as the teacher and the local ViT-S/16 as the student, optimizing the student to match the teacher's predictions on the target data.

The results, summarized in Table 10, reveal a fundamental distinction between the two approaches. Standard distillation is inherently limited by the capacity of the student model; the distilled ViT-S/16 achieves only 50.3% accuracy, failing to even match the original performance of the black-box teacher (55.5%). This result is expected, as KD aims to mimic the teacher's existing boundary rather than adapt it to the new domain.

In sharp contrast, BETA achieves 62.6% accuracy, significantly surpassing the original black-box model. This confirms that BETA is not a distillation process where a student mimics a fixed teacher. Instead, BETA utilizes the local model to actively *adapt* the input prompts for the black-box

model, allowing the final system to break through the performance ceiling of the original pre-trained weights.

### B.3 ZEROTH-ORDER OPTIMIZATION BASELINES

As a direct approach to adapting the visual prompt $\delta$ in a black-box setting, we evaluate several Zeroth-Order Optimization (ZOO) baselines. These derivative-free methods optimize the prompt by minimizing a fitness function, which we define as the Shannon entropy of the black-box model's predictions on the prompted input, $f(\delta) = \mathcal{H}(p_B(x + \delta))$. For a fair comparison, we configure all three ZOO methods to use 16 queries per test sample for their optimization process.

#### B.3.1 CMA-ES

As a representative ZOO method, **Covariance Matrix Adaptation Evolution Strategy (CMA-ES)** is a derivative-free algorithm used to optimize a high-dimensional visual prompt where gradients are inaccessible (Hansen & Ostermeier, 2001; Hansen et al., 2003; Niu et al., 2024; Meng et al., 2025). In each iteration, CMA-ES samples a population of candidate prompts from a multivariate normal distribution and evaluates them using the fitness function. The goal is to find a prompt, $\delta$, that minimizes this entropy, encouraging high-confidence predictions. Based on the performance of the sampled prompts, CMA-ES updates the mean and covariance matrix of the sampling distribution to guide the search towards more promising regions of the solution space.

#### B.3.2 RGF

**Random Gradient-Free (RGF)** is a ZOO method that estimates the gradient of the fitness function by sampling multiple random directions from a standard Gaussian distribution (Liu et al., 2018; Tsai et al., 2020). For a given visual prompt $\delta$, RGF approximates the gradient by averaging the function's response to small perturbations along these random directions, allowing it to descend the loss landscape without direct gradient calculations. The gradient approximation at iteration $t$ is computed as:

$$g_t(\delta_t) = \frac{1}{q} \sum_{i=1}^{q} \frac{f(\delta_t + \mu u_i) - f(\delta_t)}{\mu} u_i \tag{6}$$

where $u_i$ is a random direction vector drawn from $\mathcal{N}(0, I)$, $\mu$ is a small smoothing parameter, and $q$ is the number of directions sampled.

#### B.3.3 SPSA WITH GRADIENT CORRECTION (SPSA-GC)

To optimize the visual prompt under black-box constraints, we adopt the Simultaneous Perturbation Stochastic Approximation with Gradient Correction (SPSA-GC) algorithm, as utilized in Black-VIP (Oh et al., 2023). SPSA is a highly efficient ZOO algorithm that estimates the gradient using only two queries per iteration (Spall, 1992). Unlike RGF, which requires sampling multiple directions, SPSA perturbs the parameters in a single random direction and its opposite. The gradient approximation at iteration $t$ for a visual prompt $\delta_t$ is computed as:

$$\hat{g}_t(\delta_t) = \frac{f(\delta_t + \mu \Delta_t) - f(\delta_t - \mu \Delta_t)}{2\mu} \Delta_t \tag{7}$$

where $\Delta_t$ is a random perturbation vector drawn from a Bernoulli distribution, and $\mu$ is a small step size.

**Gradient Correction.** While standard SPSA is query-efficient, the stochastic gradient estimate $\hat{g}_t$ can be noisy. To mitigate this, we employ the Gradient Correction mechanism proposed in Black-VIP (Oh et al., 2023). This method integrates Nesterov's Accelerated Gradient (NAG) into the update rule, using a momentum accumulator to rectify the estimated gradient direction. By smoothing the optimization trajectory, SPSA-GC significantly enhances stability compared to vanilla SPSA, making it particularly suitable for the high-dimensional optimization of visual prompts.

Table 11: API efficiency comparison: number of API calls per test sample and performance gain.

| Method | #API Call per test sample | Accuracy (%) | Gain |
|---|---|---|---|
| Source (Inference) | 1 | 55.5 | 0 |
| LAME | 1 | 54.1 | -1.4 |
| ZOO-CMA | 16 | 54.5 | -1.0 |
| ZOO-RGF | 16 | 56.0 | +0.5 |
| ZOO-SPSA-GC | 16 | 55.1 | -0.4 |
| TTA-Aug | 64 | 55.6 | +0.1 |
| DDA | 2 | 56.9 | +1.4 |
| BETA | 1 | 62.6 | +7.1 |

### B.3.4 API EFFICIENCY COMPARISON ACROSS BLACK-BOX METHODS

Table 11 demonstrates BETA's superior efficiency compared to existing black-box TTA methods. While ZOO-based approaches (CMA, RGF, SPSA) require 16 API calls per test sample and achieve modest or negative performance gains ranging from -1.0% to +0.5%, BETA achieves a substantial +7.1% improvement with only a single API call per sample. This represents a 16× reduction in API usage while delivering significantly better adaptation performance. LAME, though equally efficient with one API call, suffers from limited adaptive capacity due to its post-hoc output refinement approach, resulting in a -1.4% performance drop. These results highlight BETA's unique combination of query efficiency and adaptation effectiveness in the black-box setting.

### B.3.5 ORTHOGONALITY OF CONTRIBUTION: UNSUPERVISED OBJECTIVE VS. ZOO ALGORITHMS

While we adopt the powerful ZOO algorithm like SPSA-GC (Oh et al., 2023) due to its superior efficiency, it is crucial to distinguish the role of the *ZOO algorithm* from the challenges inherent to the *adaptation objective*. The efficacy of SPSA-GC was originally demonstrated in BlackVIP (Oh et al., 2023) within a *supervised* few-shot transfer setting. In that context, the loss landscape is anchored by ground-truth labels via a Cross-Entropy loss, providing a consistent and convex directional signal for the zeroth-order estimator.

In contrast, our strictly **unsupervised online setting** relies on objectives such as entropy minimization. We observe that replacing the supervised loss with an unsupervised one fundamentally alters the optimization landscape, making it prone to trivial solutions. As evidenced in our experimental results, naively applying even a robust ZOO algorithm like SPSA-GC to this unsupervised objective leads to prompt collapse, where the model exploits high-frequency patterns to minimize entropy without preserving semantic integrity. Therefore, we clarify that our primary contribution does not lie in the ZOO algorithm itself. Rather, our contribution is the **unsupervised stabilization framework**: comprising Prediction Harmonization, the Coordinator architecture, and Consistency Regularization. These mechanisms effectively constrain the optimization space, preventing the instability inherent to source-free black-box adaptation and enabling effective Test-Time Adaptation.

### B.4 ROBUSTNESS TO LABEL IMBALANCE AND CONTINUAL SHIFTS

While our primary evaluation follows the standard episodic adaptation setting, real-world data streams often exhibit temporal correlations or non-stationary distributions. To validate the stability of BETA in dynamic environments, we extend our evaluation on ImageNet-C (using ViT-B/16) to include two challenging scenarios:

- **Label Imbalance** (Niu et al., 2023; Gong et al., 2022): Following the protocol established in SAR (Niu et al., 2023), we evaluate performance on data streams with highly skewed class distributions within each batch, simulating non-i.i.d. test streams.

- **Continual Domain Shifts** (Wang et al., 2022; Niu et al., 2022): Following the Continual Test-Time Adaptation (CoTTA) setting (Wang et al., 2022), the model adapts to the 15

Table 12: Robustness analysis on ImageNet-C (ViT-B/16) under Label Imbalance and Continual Domain Shift settings. BETA demonstrates minimal degradation compared to the standard setting, highlighting its stability in dynamic environments.

| Method | Gauss. | Shot | Impul. | Defoc. | Glass | Motion | Zoom | Snow | Frost | Fog | Bright. | Contr. | Elastic | Pixel. | JPEG | Avg. |
|---|---|---|---|---|---|---|---|---|---|---|---|---|---|---|---|---|
| Source | 56.8 | 56.8 | 57.5 | 46.9 | 35.6 | 53.1 | 44.8 | 62.2 | 62.5 | 65.7 | 77.7 | 32.6 | 46.0 | 67.0 | 67.6 | 55.5 |
| BETA (Standard) | 60.5 | 60.7 | 61.1 | 54.5 | 52.2 | 59.9 | 56.3 | 63.6 | 64.7 | 66.1 | 78.1 | 53.4 | 62.1 | 73.3 | 72.0 | 62.6 |
| BETA (**Label Imbalance**) | 59.0 | 59.9 | 59.5 | 53.9 | 51.1 | 59.1 | 55.5 | 62.9 | 64.3 | 65.4 | 77.9 | 52.4 | 61.2 | 73.1 | 72.1 | 61.8 |
| BETA (**Continual Shifts**) | 59.5 | 61.0 | 60.4 | 52.3 | 51.4 | 58.4 | 55.2 | 61.8 | 63.3 | 63.8 | 77.4 | 51.8 | 61.7 | 72.5 | 71.3 | 61.5 |

corruption domains of ImageNet-C sequentially without resetting the model state between domains.

The results are summarized in Table 12. BETA exhibits remarkable stability, maintaining high performance even under these challenging conditions. In the label imbalance setting, BETA achieves an average accuracy of 61.8%, and under continual shifts, it maintains 61.5%. This represents minimal degradation compared to the standard i.i.d. setting (62.6%).

**Why is BETA robust?** This robustness is intuitive given our framework's design. Unlike white-box methods that directly update internal model parameters—a process known to risk catastrophic forgetting or overfitting to biased batches—BETA keeps the parameters of the black-box target model frozen. We exclusively learn an additive input prompt. Furthermore, the local steering model is updated with a conservative learning rate and strong consistency regularization, preventing the optimization trajectory from over-fitting to the dynamic changes or local biases in the data stream. This makes BETA naturally resilient to the instability often observed in dynamic test-time adaptation.

## B.5 MORE ABLATION STUDIES

### B.5.1 ANALYSIS ON STABILIZATION MECHANISMS

We conduct a component analysis to demonstrate the importance of our two stabilization mechanisms, visualizing the online batch accuracy on the challenging ImageNet-C Contrast domain. The figure shows that the full BETA framework ("Ours") rapidly achieves high accuracy and maintains stable performance across all 800 online batches. In contrast, removing the data filtering component ("w/o Data Filtering") results in significantly lower and gradually decaying performance. More critically, removing the consistency regularization ("w/o KL Reg.") leads to catastrophic collapse, with the model's accuracy plummeting to near zero after approximately 400 batches. This analysis empirically validates that both the consistency regularization and the data filtering are essential for the stable and effective performance of BETA.

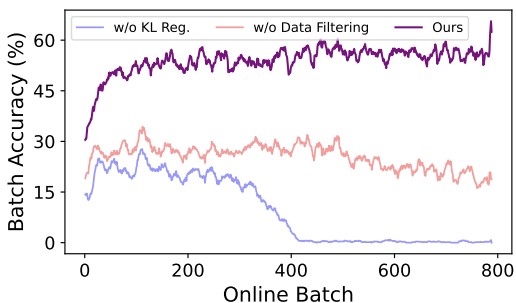

Figure 6: Online Batch Accuracy on ImageNet-C Contrast domain.

### B.5.2 ROBUSTNESS TO BATCH SIZE

In practical online deployment, the number of samples available for adaptation at any given time step can vary significantly. To assess BETA's sensitivity to this factor, we evaluated its performance on ImageNet-C (ViT-B/16) using batch sizes ranging from 4 to 128. As shown in Table 13, BETA demonstrates high robustness to batch size variations. Even with a very small batch size of 4, where gradient estimates are typically noisy, BETA achieves an average accuracy of 59.3%, significantly outperforming the source model baseline of 55.5%. The performance consistently improves as the

Table 13: Effect of Batch Size on Average Accuracy (%) on ImageNet-C. BETA consistently improves upon the Source model (55.5%) even when restricted to extremely small batch sizes.

| Batch Size | Source | 4 | 8 | 16 | 32 | 64 | 128 |
|---|---|---|---|---|---|---|---|
| Avg. Accuracy | 55.5 | 59.3 | 60.1 | 62.3 | 62.5 | 62.6 | 62.6 |

Table 14: Computational efficiency analysis on ImageNet-C (ViT-B/16). Comparison of API calls per image, local GPU memory usage, wall-clock time per image, and accuracy. BETA achieves superior performance with minimal latency, matching the speed of standard inference.

| Method | # API Calls (per image) | Local Compute Required? | GPU Mem (MB) | Time/Img (s) | Avg. Acc (%) | Gain (%) |
|---|---|---|---|---|---|---|
| Source | 1 | ✗ | - | 0.045 | 55.5 | - |
| LAME | 1 | ✓ | 2 | 0.046 | 54.1 | -1.4 |
| ZOO-SPSA-GC | 16 | ✓ | 52 | 0.450 | 55.1 | -0.4 |
| TTA-Aug | 64 | ✓ | - | 1.800 | 55.6 | +0.1 |
| DDA | 2 | ✓ | 23,427 | 12.722 | 56.9 | +1.4 |
| BETA (w/ ViT-Tiny) | 1 | ✓ | 1,292 | 0.047 | 58.2 | +2.7 |
| **BETA (w/ ViT-Small)** | **1** | ✓ | **2,616** | **0.048** | **62.6** | **+7.1** |

batch size increases, saturating at 62.6% for batch sizes of 64 and above. This indicates that while larger batches provide more stable gradients, BETA remains effective even in low-data regimes.

## B.6 Computational Efficiency and Real-Time Adaptation

To comprehensively assess the practicality of BETA, we analyze efficiency across two dimensions: API costs (query complexity) and local computational overhead. We further validate performance under a strict real-time streaming protocol, following (Alfarra et al.).

**Detailed Efficiency Breakdown.** We conducted a granular breakdown of wall-clock latency and resource usage using a single NVIDIA RTX 3090 GPU. As summarized in Table 14, we compare BETA against baselines including ZOO-SPSA-GC and Test-Time Augmentation (TTA-Aug) (Shanmugam et al., 2021).

The analysis yields two critical insights. First, **local computation is negligible** compared to API latency. While BETA introduces a local steering model (ViT-Small), it requires only 2.6GB of GPU memory—feasible for consumer-grade hardware—and adds a trivial 0.003s overhead per image for the backward pass. The primary bottleneck in black-box adaptation is the API forward pass ($T_{API} \approx 0.045s$), which is dominated by network latency. Second, **API calls dominate total latency**. Methods relying on multiple queries per image suffer from severe slowdowns. ZOO (16 calls) and TTA-Aug (64 calls) are approximately $9.4\times$ ($0.450s$) and $37.5\times$ ($1.800s$) slower than BETA per image, respectively. This clarifies the context for "backpropagation-free" approaches in this setting: eliminating the local backward pass ($0.003s$) provides no practical speed benefit when the total time is dictated by the mandatory API call ($0.045s$).

**Computationally Constrained Evaluation.** To further rigorously test feasibility in streaming scenarios, we adopt the *Realistic Evaluation Protocol* from (Alfarra et al.). This protocol penalizes methods that cannot keep pace with a data stream arriving at the API's maximum throughput speed ($r = 1 \text{ img}/T_{API}$).

We define the relative adaptation cost based on the total processing time per step: $T_{Step} = \max(T_{API}, T_{Local\_Fwd}) + T_{Local\_Bwd}$. Crucially, BETA allows for the parallelization of the local steering model's forward pass with the API query latency. Since $T_{API} \gg T_{Local\_Fwd}$, the local forward cost is effectively hidden, leaving only the negligible backward pass. Consequently, BETA maintains a relative cost $\mathcal{C} \approx 1$, allowing it to adapt to virtually 100% of the data stream. In contrast, query-intensive methods like ZOO incur massive adaptation lag ($\mathcal{C} \gg 1$), forcing them to skip adaptation for the majority of samples to maintain throughput.

Table 15: Evaluation under Computational Time Constraints (Alfarra et al.). "Offline Acc" assumes unlimited time, while "Online Acc" simulates a realistic stream where slow methods must skip samples. BETA maintains performance due to its single-query efficiency.

| Method | Offline Acc (%) | Online Acc (%) |
|---|---|---|
| Source | 55.5 | 55.5 |
| LAME | 54.1 | 54.1 |
| ZOO | 56.0 | 54.3 |
| **BETA (Ours)** | **62.6** | **62.5** |

The results in Table 15 demonstrate the impact of this constraint. Under strict real-time conditions, ZOO's performance drops to 54.3% (worse than the Source), as it updates too infrequently. BETA, however, maintains an accuracy of 62.5%, confirming it is a viable solution for real-time black-box adaptation.

## C  USE OF LARGE LANGUAGE MODELS

We used a Large Language Model to assist with language polishing and improving the readability of this manuscript. The authors are fully responsible for all research ideas, experimental results, and claims presented in this paper.

## D  LIMITATIONS

While BETA demonstrates strong performance and efficiency, its effectiveness is connected to the choice of the local steering model. In the current landscape, where most large-scale models are Transformer-based, our method is highly applicable, as finding a steering model with a similar architecture is straightforward. However, the performance could be suboptimal if the architectures of the steering and target models differ significantly. Although our experiments show that cross-architecture adaptation is effective (e.g., a CNN steering a Transformer), the improvements are slightly less pronounced than when using architecturally similar models. Another avenue for future research is extending this framework beyond classification to more versatile, generative tasks. Investigating how to adapt the harmonized objective for generative outputs, where the prediction space is vast and unstructured, would be a valuable next step.

