# OpenReview forum: "Adapting in the Dark: Towards Stable and Efficient Black-Box Test-Time Adaptation"
_ICLR.cc/2026/Conference — Submitted to ICLR 2026_

### Official Review · Reviewer_BkkH · 2025-10-16

**Soundness:** 1
**Presentation:** 1
**Contribution:** 2
**Rating:** 2
**Confidence:** 4

**Summary:**

This paper presents a black-box test-time adaptation method motivated by black-box visual prompting literature. The proposed method leverages a white-box surrogate steering model to produce a harmonized prediction with a black-box target model, and leverages the gradient of this steering model to update the visual prompt and its layer-normalization parameter. The authors propose two additional mechanisms -- input consistency (prompted image v.s. clean image) regularization and confidence-based data filtering -- to stabilize the training. They demonstrate the proposed method, BETA, on ImageNet distribution shifts benchmarks with VIT and CLIP-VIT.

**Strengths:**

* `problem definition`
  * The authors are targeting a very important, promising, yet under-explored problem -- _black-box test-time adaptation_
  * Many frontier models do not reveal their parameters, and the authors' problem setup, assuming almost completely black-box (only the logit-accessibility), is very close to the real-world foundation model adaptation setup. I believe the problem they bring and want to address is one of the contributions of this work.
* `effectiveness of the solution`
  * The proposed method shows a somewhat impressive performance gain across diverse experiments, and sometimes beats the white-box method.
  * Although not sure about this due to largely ambiguous technical details, the proposed method seems like a query-efficient way to achieve such improvements.

**Weaknesses:**

* `weak rationale and logical gap`
  * The authors do not successfully justify the use of _prediction harmonization with asymmetric optimization_ and even _prediction harmonization_ at all. The results of Figure 2 are trivial because the $g_{\text{Ideal}}$ is derived by the interpolation of $g_{\text{BETA}}$ and $g_{\text{Black}}$, and here: (1) as the green line-- Obj. Relevance--is computed between $g_{\text{Ideal}}$ and $g_{\text{Black}}$, it does not support anything about $g_{\text{BETA}}$; (2) The gold gradient is $g_{\text{Black}}$ rather than $g_{\text{Ideal}}$, so the closeness between $g_{\text{BETA}}$ and $g_{\text{Ideal}}$ does not explain why BETA will achieve a desired gradient.
  * There is a logical gap between the authors' suggestion of _consistency regularization_ and its justification. Figure 3 does not tell us the necessity of the KL regularization between the clean input and the prompted input. It just tells us how unstable the optimizations are across different runs. Why do the authors claim that this is due to the prompt collapse? And why does the consistency regularization address this?
* `missing formulation details and technical details`
  * What do they mean by $\mathcal{H(p_{x};f_{x})}$ in L214-218? they only provide definition of $\mathcal{H(p_{x})}$ in L193.
  * How many test samples (before and after filtering) were leveraged to learn your model?
  * How many epochs or iterations did you run for the optimization? This is very important for assessing the efficiency of the black-box methods.
* `naive baseline implementation`
  * The considered implementations of black-box baselines, ZOO (CMA, RGF, SPSA), are too naive. It looks like just an adoption of ZOO for visual prompt learning (Bahng et al. 2022). They do not consider advanced implementations of black-box visual prompt learning that are already available (Oh et al. 2023; Park et al., 2024).
  * This makes it hard to assess the necessity of BETA's exclusive components -- the surrogate steering model and two stabilizing mechanisms.
* `improper comparison in the number of API calls`
  * The authors' statement "the existing ZOO method requires 16 API calls per sample, while 1 API call per sample for our method". However, I believe this is an improper statement given the difference in settings of black-box visual prompting and this work.
  * The ZOO-based visual prompt learning method, e.g., BlackVIP (Oh et al. 2023) and ZIP (Park et al. 2024), was proposed in a few-shot classification setup where the number of samples required per task is $K\times C$ for the K-shot and C-class, where they set K as 16. Therefore, if C is 10, e.g., CIFAR-10, then the required number of samples for the prompt learning is 160 in total. Then, the learned prompt (or reparameterized prompt like BlackVIP) is used for all the upcoming test samples. => It does not mean it requires 16 API calls per sample.
  * Meanwhile, the proposed method BETA also learns a visual prompt from all the test samples (maybe tens of thousands or more for ImageNet variants). Therefore, BETA's prompt learning requires far more samples to learn a prompt than the existing ZOO method, and **the total number of API calls should be compared in terms of the total number of forward passes of the black-box model rather than per-sample API requirement.**

---

> Reference
- Bahng et al. 2022, "Exploring Visual Prompts for Adapting Large-Scale Models"
- Oh et al. 2023, "BlackVIP: Black-Box Visual Prompting for Robust Transfer Learning"
- Park et al., 2024, "ZIP: An Efficient Zeroth-order Prompt Tuning for Black-box Vision-Language Models"

**Questions:**

* Please clarify the technical and formulation details mentioned above.

---

### If there is any misunderstanding from me, please feel free to point it out. I would be happy to discuss.

---

> ### Author Response · Authors · 2025-11-21
> **Response to Reviewer BkkH [1/4]**
>
> We sincerely thank Reviewer BkkH for their detailed review and for recognizing the importance of the under-explored black-box TTA problem ("**important, promising, yet under-explored problem definition**") as well as the "**impressive performance gain**" of our method.
>
> ## **Clarification on Problem Setting: Online TTA vs. Few-Shot Transfer**
> ---
> We are particularly grateful for the reviewer's constructive attitude and their explicit openness to discuss. We believe the concerns regarding baselines and efficiency stem from viewing our method through the lens of **Offline Supervised Few-Shot Transfer Learning** rather than our actual setting, **Online Unsupervised Test-Time Adaptation (TTA)**. We clarify the critical distinctions below.
>
> ### 1. Setting Clarification:
> ---
> The reviewer's suggestions (e.g., BlackVIP [1], ZIP [2]) fall into the category of **Offline Few-Shot Transfer**. We formally distinguish this from our **Online TTA** setting:
>
> - **Few-Shot Transfer (e.g., BlackVIP [1], ZIP [2])**: These methods are **offline** and **supervised**. They require a labeled support set (e.g., 16 labeled images per class) to train a prompt *before* testing begins.
> - **Our Setting (Online TTA)**: We operate in the **Source-Free Online TTA** setting:
>     - **No Labels**: We have zero access to ground-truth labels.
>     - **On-the-Fly**: We do not iterate through the dataset to "learn a prompt" beforehand. The model adapts continuously during inference on the incoming test stream. Optimization occurs simultaneously with prediction.
>
> | Feature | BlackVIP [1] / ZIP [2] <br>(Few-Shot Transfer) | Our BETA <br>(Source-Free Online TTA) |
> |---------|-------------------------------------|-------------------------------|
> | Objective | Learn a *fixed* prompt offline using labels | Adapt prompt online to unlabeled test stream |
> | Data Requirement | **Labeled Support Set** (e.g., 16 shots per class) | **Unlabeled Test Data** (0 shots) |
> | Example **Training** Cost (ImageNet-Sketch) |  $\gg$ 16,000+ API calls (Labeled)  | **0** API calls  |
> | Example **Inference** Cost (ImageNet-Sketch) | 1 call/sample | 1 call/sample |
> | Mechanism | Train once, freeze, then deploy | Continuously update on incoming unlabeled batch |
>
>
> **A Concrete Example (CLIP on ImageNet-Sketch)**: To illustrate the practical difference, one example is our experiments that adapt a black-box CLIP model to ImageNet-Sketch (1,000 classes):
> - **BlackVIP/ZIP (Few-Shot Transfer)**: To work, these methods first require a **labeled training phase**. For a standard 16-shot setting, this means collecting **16,000 labeled images** ($16 \times 1000$). They use ZOO to learn a prompt on this massive labeled set before any testing begins. This incurs a massive upfront cost just to prepare the prompt. Only after this expensive training is the fixed prompt used for test data (which still requires API calls).
> - **BETA (Online TTA)**: In contrast, BETA requires **NO labeled data** and **NO offline training**. We start inference immediately. For every incoming batch of *unlabeled* test data, BETA learns and updates the prompt **on-the-fly**.
>
> This distinction is crucial: supervised methods like BlackVIP are inapplicable to our label-free setting, and our "per-sample" efficiency metric is the standard for online TTA.
>
> [1] BlackVIP: Black-Box Visual Prompting for Robust Transfer Learning, CVPR 2023
>
> [2] ZIP: An Efficient Zeroth-order Prompt Tuning for Black-box Vision-Language Models, ICLR 2025.
>
> ### 2. Access Clarification: Probabilities vs. Logits
> ---
> The reviewer characterizes our setting as "almost completely black-box" and "logit-accessibility." We respectfully clarify that our setting is **strictly Black-Box API access**.
>
> - **The "Normal User" Perspective**: We assume the role of a standard user accessing a commercial API. In this real-world setting, users have no privileged access.
> - **No Logits**: We do not assume access to raw logits. Real-world commercial APIs (like the Clarifai API we tested) typically provide only the final prediction **probabilities** (softmax outputs), not raw logits.
> - **Strict Constraint**: Methods that require logits (categorized as **gray-box** in Table 1) violate these strict API protocols. BETA operates solely on the output probabilities provided by the standard API interface.
>
> With these settings clarified, we address the specific weaknesses below.

---

> ### Author Response · Authors · 2025-11-21
> **Response to Reviewer BkkH [2/4]**
>
> ## **W1: Rationale of Prediction Harmonization & Consistency Regularization**
> ---
> ### 1. Rationale of Prediction Harmonization
> ---
> We clarify the logic behind our gradient approximation step-by-step. We have revised the manuscript (**Lines 248-286**) to explicitly formalize this derivation.
>
> 1.  **The Goal:** Our primary objective is to minimize the entropy of the black-box model ($f_B$). Ideally, we would follow the gradient $g_{Black} = \nabla \mathcal{H}(p_B)$.
> 2.  **The Problem:** We cannot compute $g_{Black}$ directly because $f_B$ is a black box. Furthermore, our analysis shows that *direct estimation* methods fail: ZOO is inefficient/unstable, and naive local gradients ($\nabla \mathcal{H}(p_S)$) have near-zero correlation with $g_{Black}$ (as shown by the gray line in Fig. 2).
> 3.  **The Relaxation (Harmonization):** To overcome this, we relax the problem to finding a prompt that improves *both* models simultaneously. We define a **Harmonized Prediction** $p_H$ (Eq. 1). The "Ideal" gradient for this shared objective—if we had full access—would be $g_{Ideal} = \nabla \mathcal{H}(p_H; f_S, f_B)$, where gradients flow through both models.
> 4.  **The Practical Proxy:** Since we cannot backpropagate through $f_B$, we approximate this ideal update by computing the gradient of the *same harmonized objective* using *only* the steering model's pathway. This yields our practical gradient: $g_{BETA} = \nabla \mathcal{H}(p_H; f_S)$.
>
> **Addressing the Notation Question (Revised in Line 250-252):** This logic directly answers the reviewer's question regarding the notation in lines 214-218.
> * $g_{Ideal}= \nabla \mathcal{H}(p_H; f_S, f_B)$ refers to the gradient of the harmonized entropy $\mathcal{H}(p_H)$ assuming white-box access to *both* $f_B$ and $f_S$.
> * $g_{BETA}= \nabla \mathcal{H}(p_H; f_S)$ refers to the gradient of the *same* harmonized entropy $\mathcal{H}(p_H)$ computed using *only* the accessible steering model $f_S$.
> * **Figure 2** validates this approach by showing that $g_{BETA}$ is a strong proxy for $g_{Ideal}$ (Optimization Effectiveness), effectively bridging the gap to the black-box model.
>
> ### 2. Logic of Consistency Regularization
> ---
> The reviewer's question about consistency regularization highlights a critical challenge specific to the **Unsupervised** setting we operate in.
>
> * **The Risk (Degenerate Solutions):** In Source-Free Online TTA, we lack ground-truth labels. **Unsupervised entropy minimization** is inherently noisy. Unlike *supervised* methods (e.g., BlackVIP/ZIP) where collapse is typically due to optimization instability, here the objective itself is dangerous: without constraints, the optimizer can find "degenerate" solutions: prompts that simply overwrite the image content with high-frequency patterns to force the model into a single, high-confidence class (e.g., classifying all inputs as "flower"). This effectively minimizes entropy but destroys semantic accuracy.
> * **Difficulty of the Problem:** We note that solving this collapse is non-trivial. Even in white-box settings, learning an input prompt via entropy minimization is known to be highly unstable and prone to collapse, often hurting performance rather than improving it [3, 4]. Previous white-box methods [3, 4] rely on **source data statistics** (e.g., feature statistics or prototypes) to constrain the update and prevent this collapse. In our **black-box source-free** setting, we do not have access to source statistics, making the problem significantly harder.
> * **The Solution (Semantic Anchoring):** The KL regularization term $\mathcal{L}_{consist}$ addresses this by enforcing that the steering model's semantic prediction on the prompted image ($x'$) remains consistent with the clean image ($x$). This "anchors" the update, ensuring the prompt adapts the domain style without corrupting the underlying semantic content.
> * **Empirical Evidence:** **Figure 6 (Appendix A.2)** provides the empirical proof requested. It shows that without this regularization ("w/o KL Reg."), the method suffers catastrophic collapse (accuracy drops to ~0%), confirming it is a necessary stabilizer for unsupervised adaptation.
>
> [3] Test-time model adaptation with only forward passes, ICML 2024.
>
> [4] OT-VP: Optimal transport-guided visual prompting for test-time adaptation, WACV 2025.

---

> ### Author Response · Authors · 2025-11-21
> **Response to Reviewer BkkH [3/4]**
>
> ## **W2: Technical Details (Formulation & Iterations)**
> ---
> We clarify the technical implementation and operational protocols.
> ### 1. Notation Clarification
> ---
> Regarding the notation in Lines 214-218 ($g_{Ideal}$ vs. $g_{BETA}$), we have provided the detailed derivation in our response to **W1**. We have also updated the manuscript to explicitly define these terms to ensure clarity for future readers (in line 250-252).
>
> ### 2. Training Protocol (Samples & Iterations)
> ---
> The questions regarding sample counts and epochs reflect the distinction between our **Online TTA** setting and offline training. We clarify the protocol below:
> * **Online Protocol:** Consistent with standard Online TTA protocols (e.g., TENT, CoTTA), we process the test stream **one batch at a time**. The "number of samples leveraged" is simply the size of the incoming test stream itself.
> * **Filtering & API Calls**: Regarding the specific question on filtering: While our method uses a dynamic filtering strategy (Eq. 2 & 4) to select only reliable samples for the gradient update, every test sample (filtered or not) requires an API call for the initial inference to generate predictions. Thus, filtering affects the update quality, not the API cost (which remains 1 call per sample).
> * **Iterations & Epochs**: We perform **one single gradient update step** per batch. There is **no offline training** and only **1 epoch** (a single pass through the stream); the model sees each test sample **only once**.
>
> This confirms the efficiency of our method: optimization happens *during* inference with no offline overhead.
>
>
> ## **W3: Baseline Selection (ZOO vs. BlackVIP/ZIP)**
> ---
> The reviewer suggests comparing against BlackVIP [1] and ZIP [2]. We clarify why these are technically inapplicable to our setting:
>
> ### 1. Inapplicability of BlackVIP and ZIP (Supervised vs. Unsupervised)
> ---
> * **BlackVIP & ZIP are Supervised:** Both methods are designed for **Few-Shot Transfer**. They fundamentally require a **labeled support set** to construct their loss functions. For instance, BlackVIP minimizes a supervised Cross-Entropy loss on labeled data to learn the prompt.
> * **BETA is Unsupervised:** In our Source-Free Test-Time Adaptation setting, **no ground-truth labels are available**. We cannot compute the supervised losses required by BlackVIP or ZIP.
> ### 2. ZOO is the Correct Unsupervised Baseline
> ---
> Since we lack labels, any black-box optimization method must rely on **unsupervised** signals (like entropy minimization).
> Our ZOO baselines adapt the "visual prompting" concept to this unsupervised setting by replacing the supervised loss with an entropy loss. This represents the standard for black-box optimization in this context.
> ### 3. Additional Baselines Added
> ---
> While BlackVIP/ZIP are not applicable, we have strengthened our evaluation by adding other relevant unsupervised black-box baselines suggested by reviewers, including **Test-time Augmentation** and **Diffusion model-based Purification**, as detailed in our response to Reviewer 2jik **W3** and Reviewer RoQK **W2**.
>
> We hope this clarifies that our choice of baselines is dictated by the constraints of the **unsupervised** TTA problem setting.

---

> ### Author Response · Authors · 2025-11-21
> **Response to Reviewer BkkH [4/4]**
>
> ## **W4: Efficiency Metrics & API Cost Analysis**
> ---
> The reviewer suggests estimating BlackVIP's cost as "160 calls" for a 10-class dataset (like CIFAR-10 or EuroSAT) in a 16-shot setup. We respectfully note that this estimate overlooks **the iterative nature of the optimizer** used in these methods. We provide a detailed breakdown below to clarify why the actual cost is significantly higher and why "API calls per sample" is the appropriate metric for our setting.
>
> **1. Offline Training vs. Online Adaptation**. The reviewer's estimate relies on an **offline protocol** (BlackVIP), where a prompt is trained on a labelled dataset, frozen, and then deployed. Crucially, even after this expensive training, the deployed model still requires **1 API call per test sample for inference**—identical to BETA. In contrast, BETA operates in the **Source-Free Online TTA** setting: there is **no offline phase** and **no labelled data**. We start inference immediately and adapt continuously to the incoming stream. Therefore, a fair comparison must account for the massive upfront "entry fee" required by offline methods, which we quantify below.
>
> **2. Analysis of API Cost (EuroSAT Case Study)**. To address the specific point about efficiency, we provide the API call estimate for the 16-shot EuroSAT (with 10 classes) scenario. BlackVIP uses Simultaneous Perturbation Stochastic Approximation (**SPSA**), an iterative optimization algorithm that requires repeated querying of the model for every image in the training set over thousands of epochs.
>
> Based on the BlackVIP [1] paper (**Appendix A.4**), the protocol is as follows:
> * **Gradient Estimation:** SPSA estimates the gradient by perturbing parameters. BlackVIP repeats this **5 times** per step to reduce variance. Each repetition requires 2 evaluations (positive and negative perturbation). This results in **10 forward passes (API calls)** for *every* image in the current batch.
> * **Training Duration:** The method trains for **5,000 epochs** for convergence.
>
> **Actual API Call Calculation (EuroSAT 16-shot):**
> * **16-shot Training Dataset:** 160 images (16 shots $\times$ 10 classes).
> * **API Calls per Epoch:** 160 images $\times$ 10 forward passes (SPSA overhead) = 1,600 API calls.
> * **Total Training Calls:** $5,000 \text{ epochs} \times 1,600 \text{ calls} = \mathbf{8,000,000 \text{ API Calls}}$.
>
> **Scaling to ImageNet (1,000 Classes)**: Crucially, this training cost scales linearly with the **number of classes**. For a large-scale dataset like ImageNet (1,000 classes), the 16-shot training set size increases by **100x** (16,000 labelled images). Consequently, the offline training cost explodes from 8 million to **~800 million** API calls.
>
> **3. Why "Online ZOO" is the Correct Baseline**
> Even if we accept the premise of comparing offline training to online adaptation, the difference in total API usage is stark in the table below (based on the 8,100 test samples of EuroSAT). In our actual **Source-Free TTA** setting, we lack the labels required to perform the offline training described above. Therefore, optimization happens at test time.
>
> | Method | Phase | # API Calls (EuroSAT) |
> | :--- | :--- | :--- |
> | **BlackVIP** (Offline) | Training (16-shot) | **~8,000,000** |
> | | Inference (Test Set) | 8,100 |
> | **ZOO-SPSA** (Online) | Training | **0** |
> | | Inference (Test Set) | 81,000 |
> | **BETA** (Online) | Training | **0** |
> | | Inference (Test Set) | 8,100 |
>
> - **BlackVIP (Different Setting)**: The massive upfront cost (~8M calls) illustrates why BlackVIP is an **Offline Few-Shot Transfer** method. It targets a different scenario where labelled data and massive offline compute are available, rather than the efficient online adaptation we propose.
> - **ZOO-SPSA (The Apple-to-Apple Comparison)**: The fair comparison for our **Online TTA** setting is **Online ZOO-SPSA**, which shares the same constraints (no labels, no offline phase).
> - **BETA’s Advantage**: Against this relevant baseline, BETA demonstrates its true value. While ZOO-SPSA requires extensive queries (**81,000 calls**) for minimal performance gain, BETA achieves superior accuracy (+11.3%) with minimal overhead (**8,100 calls**), matching the efficiency of standard inference.
>
> **4. The Necessity of Online Adaptation**
> Finally, we note that offline training is often insufficient. In real-world deployments, data distributions drift (as shown in our **Continual Shift** experiments in response to Reviewer RoQK **W3f**). A fixed, offline-trained prompt cannot adapt to changing corruptions. Online adaptation is essential for robustness, and BETA provides this capability with **zero additional API overhead**.

---

> ### Author Response · Authors · 2025-11-27
> **Follow-up on rebuttal**
>
> Dear Reviewer BkkH,
>
> Thank you again for your time. As the discussion period is coming to a close, we are writing to confirm that our explanation of the problem setting has effectively addressed your concerns.
>
> We clarified that while methods like BlackVIP/ZIP are highly effective for **offline few-shot transfer** (using a **labelled** training set), BETA targets a different scenario: directly adapting on the **unlabeled test data** in a strict **unsupervised online stream** setting. We also clarified the **API cost calculation** and the **logic of BETA** as a ***Unsupervised* Test-time Adaptation** method (distinct from ***Supervised* Transfer Learning**).
>
> We hope this addresses your concerns regarding the problem setting and rationale. Please let us know if you have any remaining questions; we are happy to provide further clarification.

---

> ### Comment · Reviewer_BkkH · 2025-11-28
>
> I appreciate the authors' professional rebuttal and significant efforts to enhance the draft. Very impressive indeed. Some of my concerns are addressed, and I want to raise my score accordingly, but the openreview system right now does not allow editing reviews; I don't know why.
>
> But let me clarify something and comment further:
>
> 1. Yes, I clearly notice the difference in settings between BlackVIP/ZIP and BETA.
>   * However, what I mean by mentioning BlackVIP and ZIP is that, if you do experiment with ZOO-CMA/RGF/SPSA, you can also do the same experiment with BlackVIP by just adopting entropy loss rather than cross-entropy (as their contribution is not a loss formulation, but the approximation of loss and a new prompt design)
>   * But I have just reminded myself that your main contribution is designing a novel unsupervised learning objective!! Indeed, this is orthogonal to the contribution of BlackVIP or similar variants, and you don't need to compare it with your vanilla method (more relevant comparison will be BlackVIP w/ entropy minimization vs. BlackVIP w/ BETA loss -- but this may be out of scope).
> 2. About the accessibility clarification, the authors said that only the single output probability value (not the logit-level access) is required, but this makes me further confused.
>   * You need to know the full output probability distribution to compute the entropy term, as it requires a C-dimensional class-wise probability distribution rather than a single probability value (probability of the argmax index across the full distribution).
>   * How is it different from the logit-level access (pre-softmax C-dimensional vector)? What I mean by this is that your method requires the output distribution across the class level rather than a single output value. Do I misunderstand something here?

---

> ### Author Response · Authors · 2025-11-28
> **Thank you for your response and intention to raise the score!**
>
> Dear Reviewer BkkH,
>
> We are thrilled to hear that our rebuttal has successfully addressed your concerns, and we sincerely thank you for considering raising your score. We deeply appreciate your engagement and your "impressive" assessment of our efforts.
>
> ### 1. Clarification on Optimization Algorithm vs. Unsupervised Adaptation Objective
>
> We fully agree with your insight. Our core contribution lies in the unsupervised learning objective itself (for online adaptation), which addresses challenges distinct from the optimization algorithm.
>
> To be precise, the "ZOO-SPSA" baseline in our experiments is implemented utilizing the **"VP w/ SPSA-GC"** algorithm from BlackVIP, as we now clarify in **Appendix Sec.B.3.3** of the updated manuscript. We explicitly selected this implementation because SPSA-GC represents a more recent and powerful zeroth-order optimization technique compared to standard SPSA. As our empirical results demonstrate, the primary difficulty does not lie in the ZOO algorithm itself, which is highly effective in supervised transfer learning, but rather in the fact that replacing the supervised cross-entropy loss (used in BlackVIP) with an unsupervised entropy loss significantly exacerbates instability.
>
> Therefore, we align with your view: our contribution does not lie in the optimization algorithm. Instead, our contribution is the **unsupervised objective and stabilization mechanisms** (Harmonization + Consistency) and the **query-efficient design** (single API call) that enables robust black-box adaptation when labels are unavailable and real-time efficiency is required. Following your insightful suggestion, we have updated the manuscript to explicitly discuss this orthogonality in **Appendix Sec.B.3.5**.
>
>
>
> ### 2. Clarification on "Logits vs. Probabilities"
>
> We appreciate the opportunity to clarify the exact level of access. To be precise, we define the terms as follows:
>
> - **Raw Logits:** The pre-softmax $C$-dimensional vector (unbounded values).
> - **Softmax Probability Vector:** The post-softmax $C$-dimensional output distribution (normalized values summing to 1).
> - **Top-1 Probability:** A single scalar value (the largest probability).
>
> **Our Setting (Real-World Black-Box):** Real-world commercial APIs (e.g., Clarifai, OpenAI) typically return the **Softmax Probability Vector**. They strictly hide **Raw Logits** to prevent model distillation and ensure safety. Therefore, we define the "Black-Box" setting as having access to the Softmax Probability Vector, consistent with these real-world constraints.
>
> **BETA’s Access:** Yes, BETA utilizes the full $C$-dimensional Softmax Probability Vector, not just a single Top-1 scalar value, to compute the entropy.
>
> **Distinction from Gray-Box:** We categorize methods that require **Raw Logits** as "Gray-Box" (e.g., ZERO [a], which requires logits to perform softmax with different temperatures). Since standard APIs do not provide raw logits, such methods are often inapplicable in strict API scenarios. In contrast, BETA operates strictly within the constraints of standard API outputs (Post-Softmax). We have added a detailed analysis of these access levels and security constraints in **Appendix A.2** of the updated paper.
>
> Thank you again for your constructive feedback and support! If you have any remaining concerns, please do not hesitate to let us know; we are happy to provide further clarification.
>
> ---
> [a] Frustratingly Easy Test-Time Adaptation of Vision-Language Models, NeurIPS 2024.

---

### Official Review · Reviewer_RoQK · 2025-10-30

**Soundness:** 4
**Presentation:** 2
**Contribution:** 3
**Rating:** 6
**Confidence:** 4

**Summary:**

This paper proposes a Test-Time Adaptation method (TTA) that is applicable to black-box setting. Through a steering model, and optimizing a visual prompt, the proposed BETA operates and adapts the prediction of black-box models and improves their performance under distribution shifts.
Experiments are carried out on standard ImageNet-C/S/R benchmarks showing consistent performance gain.

**Strengths:**

The main strengths of this work are:

1) This paper studies a unique setting: adaptation under black box setting. This is very useful for many applications in the era of strong models deployed as APIs

2) The proposed BETA is simple, cost efficient, and effective

3) Experiments are thorough and results are convincing

**Weaknesses:**

Generally, there are no major weaknesses in my opinion for this paper. The following points are suggestions to make the argument stronger:

1) The writing of this paper can be vastly improved. Here are a list of suggestions to make the narrative more convincing and easier to follow:

1a) There is no clear mention why should we study TTA under the black-box setting as generally adaptation is a responsibility of the model provider, thus having full white-box access. One possible argument that can be added in the introduction is developing stronger inference of the already strong API at the user side.

1b) Table 1 is not referenced in the text.

1c) Figure 1 can be aesthetically improved significantly.

1d) It is not completely clear in Figure 2 why considering $\alpha \in [0.4, 0.6]$ is optimal. The text accompanying that figure describing the experiment should be revised and improved for better clarity.

2) While the experiments considered several baselines, there are two relevant competitors that went missed. Both DDA [A] and DiffPure [B] operate in the black box setting. Particularly, DDA denoises the received input with added perturbation (through diffusion process), in a slightly similar way to BETA. I would highly recommend including experimental comparison against them, especially [A]

3) There are some experimental ablations that are interesting to show and could complement the results in the paper:

3a) Does the learnt visual prompt have any semantic information? Does it denies the received corrupted image?

3b) The cost argument comparing BETA with ZO methods is convincing. One possible standardized way to compare these two schemes is through the computational constrained evaluation [C].

3d) While SAR is included in the comparison tables, ETA/EATA [D] are not. Any reason why?

3f) All experiments in the main paper are conducted under the episodic evaluation setting. I would also recommend including evaluations under the continual / label imbalanced settings [E].

[A] Back to the source: Diffusion-driven test-time adaptation, CVPR 2023

[B] Diffusion Models for Adversarial Purification, ICML 2022

[C] Evaluation of Test-Time Adaptation Under Computational Time Constraints, ICML 2024

[D] Efficient test-time model adaptation with- out forgetting, ICML 2022

[E] Robust test-time adaptation in dynamic scenarios, CVPR 2023

I am happy to raise my score if the aforementioned points are addressed.

**Questions:**

Please refer to the weaknesses section

---

> ### Author Response · Authors · 2025-11-21
> **Response to Reviewer RoQK [1/3]**
>
> We sincerely thank Reviewer RoQK for their constructive feedback and for recognizing the value of our work. We are encouraged that the reviewer found BETA to be "**simple, cost efficient, and effective**," assessing our results as "**convincing**." We appreciate the insightful suggestions to strengthen our narrative and baselines, which we have addressed below.
>
> ## **W1: Writing and Presentation Improvements**
> ---
> We have revised the manuscript to address all presentation points:
>
> - 1a) **User-Centric Motivation**: We updated the Introduction to explicitly articulate the motivation from the user's perspective (**Lines 37-39**), clarifying that TTA empowers users to develop stronger inference capabilities for fixed APIs.
> - 1b: **Table 1 Reference**: We added an explicit reference to Table 1 in the main text to guide the comparison (**Line 53**).
> - 1c: **Figure 1 Aesthetics**: We acknowledge the feedback and will improve the aesthetic quality of Figure 1 in the camera-ready version.
> - 1d) **Figure 2 Clarification**: We revised the text and caption to clearly explain that the optimal $\alpha$ strikes a balance between Objective Relevance and Optimization Effectiveness (**Lines 249-286**).
>
>
> ## **W2: Comparison with Diffusion-Based Baselines (DDA  [A] and DiffPure [B])**
> ---
> We thank the reviewer for suggesting these relevant works. We implemented DDA [A] as an additional baseline. Our analysis shows that BETA significantly outperforms diffusion methods in efficiency, applicability, and robustness.
>
> 1. **Applicability in Black-Box Settings**
>     - **Data Requirement**: DDA requires **training a diffusion model on the source data**. In our target setting (e.g., using a commercial API), the user typically does not have access to—*or even knowledge of*—the provider's private training data, making DDA inapplicable.
>     - **Domain Scope**: DDA and DiffPure struggle with **global, low-frequency shifts** (e.g., Fog) or **distinct semantic gaps** (e.g., EuroSAT). As noted in [A], diffusion models often fail to distinguish such global shifts from inherent image content, leading to reconstruction failures or hallucinations rather than useful adaptation. Consequently, they lack generality for fine-grained domain shifts that cannot be modelled simply as "noise." In contrast, BETA is a general framework that effectively handles both corruptions and complex semantic transitions (as shown in our **EuroSAT** results in Reviewer 2jik **W3**).
> 2.  **Efficiency (BETA is ~200x Faster)**. Test-time adaptation requires low latency. DDA is computationally heavy due to iterative sampling.
>     - **DDA/DiffPure**: These methods are computationally intensive due to iterative diffusion sampling. On our hardware RTX 3090, DDA takes **~10.8s** per image, and DiffPure takes **~25.4s** per image.
>     - **BETA**: In contrast, BETA takes only **~0.05s** per image (including API latency).
>     - **Conclusion**: BETA is over **200x** faster, making it viable for real-time applications where diffusion methods are prohibitive.
>
> 3. **Performance Comparison** We evaluated BETA against DDA on ImageNet-C (ViT-B/16). Due to the extreme computational cost of diffusion methods, we prioritized the stronger baseline, DDA, for this rebuttal (noting that the DDA paper itself reports DiffPure **degrades** performance on ImageNet-C relative to the source). We will include full DiffPure results in the camera-ready version. As the table below shows, while DDA handles noise-based corruptions well, it lacks robustness across diverse global shifts. In contrast, BETA achieves consistent improvements across all domains, offering a **superior trade-off (5.7% higher performance and 200x faster)** between robustness, applicability, and efficiency.
>
> |Method|gauss.|shot|impul.|defoc.|glass|motion|zoom|snow|frost|fog|bright.|contrast|elastic|pixelate|jpeg|Avg Acc|Gain|
> |-|-|-|-|-|-|-|-|-|-|-|-|-|--|-|-|-|-|
> | Source | 56.8 | 56.8 | 57.5 | 46.9 | 35.6 | 53.1 | 44.8 | 62.2 | 62.5 | 65.7 | 77.7 | 32.6 | 46.0 | 67.0 | 67.6 | 55.5 |  |
> | DDA [A] | 64.7 | 65.0 | 64.6 | 46.3 | 41.3 | 51.7 | 43.7 | 59.1 | 61.3 | 45.0 | 74.9 | 40.6 | 54.4 | 72.2 | 68.4 | 56.9 | +1.4 |
> | BETA | 60.5 | 60.7 | 61.1 | 54.5 | 52.2 | 59.9 | 56.3 | 63.6 | 64.7 | 66.1 | 78.1 | 53.4 | 62.1 | 73.3 | 72.0 | 62.6 | +7.1 |

---

> ### Author Response · Authors · 2025-11-21
> **Response to Reviewer RoQK [2/3]**
>
> ## **W3: Ablations and Additional Comparisons**
> ---
> ### 3a) Semantic Nature of the Visual Prompt
> ---
> We analyzed the learned visual prompts ($\delta$) and found that they **do not contain explicit semantic information** (such as recognizable objects). Instead, they manifest as high-frequency, structured patterns.
>
> This observation is consistent with findings in the Visual Prompting literature ([1]), which note that visual prompts act as "visual cues" to steer the model and are "**not necessarily interpretable to humans**" . Critically, we clarify that these prompts do not simply "denoise" the image (i.e., restore original pixels). As our comparison with explicit denoising methods (DDA) in **W2** shows, pure denoising is often insufficient for complex domain shifts. Instead, the prompt acts as a **domain-adaptive perturbation**—similar to "adversarial reprogramming" ([2])—that shifts the target features into the model's reliable manifold. **We will include these visualizations in the appendix of the camera-ready version**.
>
> ### 3b) Computationally Constrained Evaluation [C]
> ---
> We adopted the **Realistic Evaluation Protocol** from [C], which penalizes adaptation methods that cannot keep pace with a constant-speed data stream. (See also our detailed **wall-clock analysis** in the response to Reviewer 2jik **W4**).
>
> **1. Definitions & Setup**:
> - $T_{API}$: Latency of a single image query to the black-box API ($\approx 0.045s$).
> - $T_{Local\_Fwd}$ / $T_{Local\_Bwd}$: Computation time for the local steering model's forward and backward passes, respectively ($T_{Local\_Bwd} \approx 0.003s$).
> - **Stream Speed**: We set the data arrival rate to match the API's maximum throughput ($r = 1 \text{ img} / T_{API}$), simulating a strict real-time limit.
> - **Latency Model**: The total processing time per step is $T_{Step} = \max(T_{API}, T_{Local\_Fwd}) + T_{Local\_Bwd}$.
>
> **2. Relative Adaptation Cost ($\mathcal{C}$)**:
> - **BETA ($\mathcal{C} \approx 1$)**: Crucially, BETA **parallelizes** the local steering model's forward pass ($T_{Local\_Fwd}$) with the API query. Since API latency dominates ($T_{API} \gg T_{Local\_Fwd}$), the forward pass cost is hidden. The only serial cost is the negligible backward pass ($T_{Local\_Bwd} \approx 0.003s$). Thus, BETA adapts to virtually 100% of the stream.
> - **ZOO ($\mathcal{C} \gg 1$)**: ZOO requires $K$ sequential API queries to estimate one gradient (typically $K \approx 16$). This creates massive "adaptation lag," forcing ZOO to update only once every $\sim16$ samples while skipping adaptation for the rest to maintain throughput.
>
> | Method | [C]'s Offline Acc. | [C]'s Online Acc.  |
> |--------|-------|-------|
> | Source | 55.5 | 55.5 |
> | LAME | 54.1 | 54.1 |
> | ZOO | 56.0 | 54.3 |
> | BETA | 62.6 | 62.5 |
>
>
> 3. **Results**: Under this protocol, ZOO's performance drops to **54.3%** (worse than Source), while BETA maintains **62.5%**, proving it is the only viable method for real-time black-box adaptation.

---

> ### Author Response · Authors · 2025-11-21
> **Response to Reviewer RoQK [3/3]**
>
> ### 3d) Comparison with ETA/EATA [D]
> ---
> We focused our comparisons on methods that operate in strict black-box settings. White-box methods (like SAR, TENT, and EATA) are included primarily as reference points rather than direct baselines, as they require privileged access (gradients/parameters) that is impossible in our setting. However, for completeness, we have added ETA [D] (the source-free variant of EATA) to our reference comparisons in the revised manuscript (**Table 2 and 3**) because standard EATA is often not applicable in strict source-free settings as it typically requires **Fisher Information from the source data**.
> ### 3f) Robustness to Label Imbalance and Continual Shifts
> ---
> We extended our evaluation on ImageNet-C (ViT-B/16) to include **Label Imbalance** (following SAR) and **Continual Domain Shifts** (following [D] and [3]). As the results demonstrate, BETA exhibits remarkable stability, maintaining high accuracy (61.8% and 61.5%) with minimal degradation compared to the standard setting (62.6%) even under challenging imbalance and continual shift scenarios.
>
> |Method|gauss.|shot|impul.|defoc.|glass|motion|zoom|snow|frost|fog|bright.|contrast|elastic|pixelate|jpeg|Avg Acc|
> |-|-|-|-|-|-|-|-|-|-|-|-|-|--|-|-|-|
> | Source | 56.8 | 56.8 | 57.5 | 46.9 | 35.6 | 53.1 | 44.8 | 62.2 | 62.5 | 65.7 | 77.7 | 32.6 | 46.0 | 67.0 | 67.6 | 55.5 |
> | BETA | 60.5 | 60.7 | 61.1 | 54.5 | 52.2 | 59.9 | 56.3 | 63.6 | 64.7 | 66.1 | 78.1 | 53.4 | 62.1 | 73.3 | 72.0 | 62.6 |
> | BETA (**in Label Imbalance Setting (SAR)**) | 59.0 | 59.9 | 59.5 | 53.9 | 51.1 | 59.1 | 55.5 | 62.9 | 64.3 | 65.4 | 77.9 | 52.4 | 61.2 | 73.1 | 72.1 | 61.8 |
> | BETA (**in Continual Shifts Setting [D, 3]**) | 59.5 | 61.0 | 60.4 | 52.3 | 51.4 | 58.4 | 55.2 | 61.8 | 63.3 | 63.8 | 77.4 | 51.8 | 61.7 | 72.5 | 71.3 | 61.5 |
>
> **Why is BETA robust?** As the results show, BETA maintains strong performance (61.8% and 61.5%) with minimal degradation compared to the standard setting (62.6%). This robustness is intuitive given our design:
>
> - **Frozen Target Model**: Unlike white-box methods that update model weights (risking catastrophic forgetting or overfitting to biased batches), BETA keeps the black-box model frozen.
> - **Stable Prompting**: We only learn an additive input prompt. The local steering model is updated with a small learning rate and strong regularization, preventing it from overfitting to dynamic changes in the data stream. This makes BETA naturally resilient to the instability often seen in dynamic test-time adaptation.
>
> [1] Exploring Visual Prompts for Adapting Large-Scale Models.
>
> [2] Adversarial Reprogramming of Neural Networks, ICLR 2019.
>
> [3] Continual Test-Time Domain Adaptation, CVPR 2022.

---

> > ### Comment · Reviewer_RoQK · 2025-11-25
> >
> > I would like to thank the authors for their efforts in replying to my questions. Most of my concerns are resolved, thus I am keeping my original positive score for this paper. When checking the updated version of the paper (and appendix), I did not find most of the experiments done for the rebuttal (while I do think they will strengthen the paper). Any reason why?

---

> ### Author Response · Authors · 2025-11-26
> **Response to PDF Query and Confirmation of Updated Results**
>
> Dear Reviewer RoQK,
>
> Thank you for your prompt reply. We are happy to hear that most of your concerns have been resolved. We genuinely appreciate your support and your constructive suggestion to formalize the rebuttal results in the manuscript.
>
> We fully agree that including these experimental results significantly strengthens the paper. Regarding your question on why they were not updated in the PDF:
>
> 1. **Timing and Thoroughness:** We prioritized sharing the raw results in the rebuttal text immediately to demonstrate the method's effectiveness without delay. However, we also wanted to go beyond your initial request to ensure the final manuscript was as comprehensive as possible. Specifically, to address your request for **Diffusion Purification comparisons (W2)**, we took the extra time to run the **DDA baseline on the larger ViT-L/16 architecture** (in addition to ViT-B) to prove that our efficiency and performance advantages hold across model scales.
> 2. **PDF Update:** We have just updated the PDF to formally include these results. You can now find the specific data points you requested in the following sections:
>    - **Comparison with DDA/Diffusion (W2):** We have updated **Tables 2 & 3** (Main Paper) to include DDA results on **both ViT-B/16 and ViT-L/16**. Notably, the **new results on ViT-L/16** confirm that BETA (+4.0%) consistently outperforms DDA (+1.5%) on larger architectures.
>    - **Computational Constrained Evaluation (W3b):** The full analysis using the "Realistic Evaluation Protocol" [C] is now included in **Appendix B.6**, demonstrating BETA's viability in real-time streams where ZOO fails.
>    - **Comparison with ETA (W3d):** We have added ETA [D] to the main comparison tables (**Table 2 & 3**) as a reference point.
>    - **Performance in Non-episodic Setting (W3f):** The evaluations under **Label Imbalance** and **Continual Domain Shift** settings [E] are now in **Appendix B.4 (Table 12)**.
>
> We are committed to ensuring all claims are backed by this rigorous data in the final version.
>
> We sincerely hope that the formal inclusion of these additional results in the updated PDF resolves your remaining concerns. We are truly grateful for your insightful feedback, which has played a crucial role in making our paper more rigorous and complete.
>
> Given that we have strived to address all the missing baselines, efficiency questions, and presentation issues you highlighted, **we would be incredibly grateful if you would consider raising your score**, as mentioned at the end of your initial review.
>
> Thank you very much for your time, patience, and support throughout this process.
>
> Best regards,
>
> The Authors

---

> > ### Comment · Reviewer_RoQK · 2025-11-27
> > **Thank you**
> >
> > I would like to thank the authors for their efforts. Given that my concerns have been addressed, I raise my score.

---

> > > ### Author Response · Authors · 2025-11-27
> > > **Thanks for increasing to 8!**
> > >
> > > Thank you for your time and for raising your score to 8! We appreciate the constructive feedback, which has helped significantly improve the final version of the paper.

---

### Official Review · Reviewer_2jik · 2025-11-01

**Soundness:** 2
**Presentation:** 2
**Contribution:** 3
**Rating:** 2
**Confidence:** 3

**Summary:**

The article addresses the problem of test-time adaptation in the black-box setting, i.e., where the developer has no access to the model parameters but only to the input and output, treating the model as an API. The article formalizes this setting and proposes a model for it, BETA. BETA instantiates a local steering model that is trained to minimize the entropy for the prediction of steering and the original model combined on samples with high confidence predictions from the latter. A visual additive prompt is also tuned using consistency on the original model output with clean inputs. Finally, the normalization layers of the steering model are also updated using a steering loss, accounting for reliability and non-redundancy of samples. Results show advantages w.r.t. alternative approaches on various benchmarks.

**Strengths:**

1. The article gives a clear formalization of the problem, with 3.1 describing the various TTA settings and their assumptions. The black-box one indeed is very challenging, and of clear importance despite the little attention given to it.

2. The analyses of Fig. 4 showing the API actual cost on a commercial platform are a very interesting experiment, clearly demonstrating the (monetary) advantages of adopting the proposed pipeline. Given that black-box models are treated as APIs, this table makes a strong point in favour of adopting the proposed approach.

**Weaknesses:**

1. The idea behind the approach (i.e., improving test-time performance using an auxiliary model) is interesting, yet it might be arguable whether we are adapting the target model or mostly distilling its knowledge into a smaller one. In particular, while Tab. 7 shows the limitations of performing TTA on the steering model, it would be interesting to show the results of simple distillation of the output from the target to the steering model, as distillation strategies have already been shown to be effective for boosting the performance of smaller models on TTA [a].

2. This might be subjective, but I found Sec. 3 quite hard to follow. The method is complex, involving several design choices (e.g., perturbing the input, learning a prompt, updating normalization layers), and the flow between the different components requires some back and forth. For instance, the model introduces the aggregated prediction score with Eq. (1), but the first loss that is introduced is the consistency loss on the predictions of the steering model. Eq. (1) and its definition are used only in Eq. (3) later, whilst being the first thing that is defined. What is updated for each objective is not always clear (i.e., the consistency updates only the input or also the normalization layers?). Changing the structure by starting with the parts where the target and steering model communicate and then moving to the various regularization objectives (a flow followed also in the introduction) might ease the understanding of the approach.

3.  The set of baselines is limited. In particular, there exist TTA methods based on augmentation strategies (e.g., [b,c]), not requiring any training or access to the parameters of the model, but just input modifications. Moreover, there are approaches tailored for black-box TTA on VLMs (i.e., Meng et al. 2025/[d]), and it is unclear why they are categorized as gray boxes (lines 150-151), as, to my knowledge, it does not assume access to internal features/tokens. Note that the VLM approaches even outperform the proposed strategy on VLM benchmarks: e.g., in Tab. 4, the proposed BETA performs 76.0 on ImageNet-S against the 77.2/80.8 of [c] and 76.6 of [d]. Enlarging the set of baselines for the black-box part (even by adapting approaches, when feasible) would strengthen the experimental results and thus the main claims in terms of effectiveness.

4. While the approach is claimed as "efficient", this refers to the number of API calls (e.g., Tab. 8) rather than the computational cost. Compared to alternatives (e.g., augmentation strategies above, Meng et al. 2025, Boudiaf et al. 2022), the approach may require more time due to the training objectives and updates. A deeper analysis (e.g., memory, speed) would be helpful to understand the tradeoff between API and computational costs of various approaches.

5. Following on the previous point, the article points to the stability of the approach, mostly referred to as the robustness ot noisy/signals that can lead to degenerate solutions (e.g., lines 70-76). However, there is no analysis on, e.g., robustness of the approach w.r.t. order of test data (e.g., i.i.d. vs non i.i.d., as in [e,f]), batch size, and/or other changes in the testing conditions. Adding these analyses would strengthen the claim of the efficacy of the approach.

6. The way the steering is initialized might have a huge impact on the performance. Specifically, while Tab. 6  demonstrates that assumptions on the architectures can be easily overcome, there is no analysis on whether the pretraining dataset between the source and target should match. This could be interesting to understand which type of assumptions w.r.t. the target model are needed to ensure the effectiveness of the framework.

7. While the motivation of the work is clear, given that VLMs are (arguably) more prone to be released as black-boxes, plus they have been the focus of various recent TTA works, it would have been interesting to extend the analysis to more settings of those for test-time adaptation of VLMs (e.g., fine-grained datasets, full set of ImageNet variants (e.g., (Manli et al., 2022) [c,d]) instead of limiting it to a single architecture and two scenarios as of Tab. 4.

**References:**

[a] Zhao, Shuai, et al. "Test-time adaptation with clip reward for zero-shot generalization in vision-language models." ICLR 2023. \
[b] Shanmugam, Divya, et al. "Better aggregation in test-time augmentation." ICCV 2021.\
[c] Farina, Matteo, et al. "Frustratingly easy test-time adaptation of vision-language models." NeurIPS 2024.\
[d] Meng, Fan'an, et al. "Black-Box Test-Time Prompt Tuning for Vision-Language Models." AAAI 2025.\
[e] Gong, Taesik, et al. "Note: Robust continual test-time adaptation against temporal correlation." NeurIPS 2022.\
[f] Niu, Shuaicheng, et al. "Towards stable test-time adaptation in dynamic wild world." ICLR 2023.

**Questions:**

1. What are the advantages of the approach vs distillation based strategies?
2. Which of existing approaches for TTA or test-time augmentation could be included as competitors? How would the model perform against them?
3. Is the model efficient in terms of computational cost?
4. Is the model robust to batch ordering and size?
5. How is the steering model initialized?

---

> ### Author Response · Authors · 2025-11-21
> **Response to Reviewer 2jik [1/4]**
>
> We thank Reviewer 2jik for their detailed and constructive feedback. We are encouraged that the reviewer appreciated the **"clear formalization of the problem"** and found our real-world API cost analysis (Figure 4) to be a **"very interesting experiment"** that makes a **"strong point"** for our method.
>
> Below, we address the concerns regarding baselines, efficiency, and stability.
>
> ## **W1 & Q1: BETA vs. Distillation (Is this just distillation?)**
> ---
> We thank the reviewer for this insightful question. To verify whether BETA is simply "distilling" the black-box model, we implemented a **Test-Time Knowledge Distillation (KD)** baseline following [a]. We distilled knowledge from the black-box teacher (ViT-B/16) into the local student (ViT-S/16) using the target data.
>
> **Results:** As shown below, standard distillation is limited by the student's capacity. The distilled student achieves only **50.3%**, failing to even match the original black-box source performance (55.5%).
>
> | Model          | Method                | Average Accuracy (%) |
> |----------------|-----------------------|----------------------|
> | ViT-S/16  (Steering Model)     | Source   | 39.5                 |
> |  | TENT  | 51.9                 |
> |                | KD (from ViT-B/16) | 50.3        |
> | ViT-B/16   (Black-Box Model)    | Source     | 55.5                 |
> |  | Ours (BETA)        | 62.6                 |
>
>
> **Conclusion:** This confirms that **BETA is not distillation.** Distillation aims to mimic the teacher on a weaker student (upper-bounded by the teacher). BETA uses the local model to **adapt the prompt for the teacher itself**, allowing the final system to significantly surpass the original black-box performance (**62.6%** vs. 55.5%).
>
> ## **W2: Method Description Clarity (Section 3)**
> ---
> We appreciate the feedback on the flow of Section 3. We have revised the manuscript to present the components in the logical order suggested by the reviewer: starting with the interaction between the target and steering models (prediction harmonization), followed by the specific regularization objectives. (please refer to **the updated manuscript**).

---

> ### Author Response · Authors · 2025-11-21
> **Response to Reviewer 2jik [2/4]**
>
> ## **W3, W7 & Q2: New Baselines and VLM Comparisons**
> ---
> We agree that a wider set of baselines strengthens our claims. We have expanded our comparisons as follows:
>
> ### 1. Augmentation-Based Baselines [b, c]
> ---
> We examined [b] and [c]:
> - Regarding **[b] (Better Aggregation)**, this is a **supervised offline** adaptation method that learns aggregation weights on a labelled dataset using a cross-entropy loss. As such, it is not applicable to our unsupervised, source-free Test-Time Adaptation setting.
> - Regarding **[c] (ZERO)**, it requires **access to raw logits** to adjust temperature, whereas our setting assumes access only to the final prediction probabilities, which is the norm for most commercial APIs.
> - **New Baselines Implementation**: Despite this, to provide a comprehensive comparison, we have:
>     1. Added **ZERO** and **ZERO_ensemble** to our VLM tables, granting them the logit access they require. These methods are, as the reviewer notes, highly query-intensive: ZERO requires 64 API calls per input, while ZERO_ensemble requires 448 API calls per input (resulting from 64 image augmentations $\times$ 7 different text templates).
>     2. Implemented a new "**Test-time Augmentation (TTA-Aug)**" baseline for our VM experiments (Table 1 below), which, similar to ZERO, uses 64 augmentations per image.
>
> ### 2. Why B²TPT [d] is "Gray-Box"
> ---
> We categorized B²TPT as gray-box because it requires modifying inputs in the **embedding/token space** (prepending vectors to $e_t$ and $e_v$, per their Eq. 3).
> Although its title suggests "Black-Box," its "Intrinsic Prompt Generation" section (and Figure 1) states that it learns prompts in the **embedding/token space**, not from raw inputs; indeed, their Eq. 3 (following TPT) shows the prompts $p_t$ and $p_v$ are directly prepended to the text embedding $e_t$ and image embedding $e_v$, respectively.
>
> This requires internal access to intermediate features, which is characteristic of a gray-box setting and is not feasible on real-world APIs that only accept **raw images and text**; furthermore, their ZOO-based approach (i.e. CMA-ES) requires 120 API calls per input (4 iterations and 30 samples per iteration). However, for a thorough comparison on VLMs, we have included its reported results in our new VLM table.
>
> ### 3. Results: BETA vs. Augmentation Baselines
> ---
> Following the reviewer's suggestion, we have tested these on ImageNet-C (VMs) and expanded our VLM experiments to the "**full set of ImageNet variants**" and also added **a new fine-grained dataset, EuroSAT**, which is noted in [c] (ZERO) as a challenging benchmark.
>
> **Key Findings (Tables below)**
> - **Performance**: BETA consistently **outperforms** all augmentation baselines.
> - **Efficiency**: Augmentation methods are expensive. ZERO_ensemble requires 448 API calls per image. At commercial rates (Clarifai: \\$0.0032/call), processing a single image costs ~\\$1.43. In contrast, BETA uses 1 call and achieves higher accuracy. Furthermore, many APIs have strict rate limits, making 448 concurrent requests for a single sample unrealistic and severely **prolonging the adaptation** process.
>
> Therefore, our method, **with only one API call per input**, is demonstrably more effective, efficient, and practical.
>
> **Results on ImageNet-C with ViT-B/16**
> |Method|gauss.|shot|impul.|defoc.|glass|motion|zoom|snow|frost|fog|bright.|contrast|elastic|pixelate|jpeg|Avg Acc|Gain|#API per Image |
> |-|-|-|-|-|-|-|-|-|-|-|-|-|--|-|-|-|-|-|
> |Source|56.8|56.8|57.5|46.9|35.6|53.1|44.8|62.2|62.5|65.7|77.7|32.6|46.0|67.0|67.6|55.5|1 |
> |LAME|56.5|56.5|57.2|46.4|34.7|52.7|44.2|58.4|61.5|63.1|77.4|24.7|44.6|66.6|67.2|54.1|-1.4|1 |
> |ZOO-SPSA|59.6|58.7|60.2|47.9|38.0|53.7|44.7|58.2|61.7|63.6|76.7|12.7|49.4|70.7|70.2|55.1|-0.5|16 |
> |TTA-Aug|55.4|54.2|55.2|43.7|48.6|48.9|45.5|57.8|63.1|60.0|76.9|49.6|41.7|65.7|67.8|55.6|+0.1|64 |
> |BETA (Ours)|60.5|60.7|61.1|54.5|52.2|59.9|56.3|63.6|64.7|66.1|78.1|53.4|62.1|73.3|72.0|62.6|+7.0|1 |
>
> **Results on ImageNet Variants with CLIP**
> |Method|ImageNet-S|ImageNet-R|ImageNet-A|ImageNet-v2|ImageNet|Average Acc|Gain|#API per Image |
> |-|-|-|-|-|-|-|-|-|
> |Source|46.1|74.0|47.9|60.9|66.7|59.1|1 |
> |LAME|45.4|72.8|48.1|61.6|66.7|58.9|-0.2|1 |
> |ZOO-SPSA|46.0|72.8|50.2|61.5|65.8|59.3|+0.1|16 |
> |B²TPT (gray-box)|49.5|78.6|55.3|65.4|69.6|63.7|+4.6|120 |
> |ZERO (with logits)|48.4|77.2|59.6|64.2|69.3|63.7|+4.6|64 |
> |ZERO_ensemble (with logits)|50.6|80.8|62.8|65.2|71.2|66.1|+7.0|448 |
> |BETA (Ours)|50.9|76.0|62.8|65.1|77.5|66.5|+7.4|1 |
>
> **Results on EuroSAT with CLIP**
> |Method|EuroSAT|Gain|#API per Image |
> |-|-|-|-|
> |Source|42.0|1 |
> |B²TPT (gray-box)|46.8|+4.8|120 |
> |ZERO (with logits)|39.6|-2.4|64 |
> |ZERO_ensemble (with logits)|43.8|+1.8|448 |
> |BETA (Ours)|53.3|+11.3|1 |

---

> ### Author Response · Authors · 2025-11-21
> **Response to Reviewer 2jik [3/4]**
>
> ## **W4 & Q3: Computational Efficiency Analysis**
> ---
> We agree that "efficiency" has two dimensions: API costs and local compute. We performed a detailed breakdown (Table below) using a single RTX 3090.
>
> | Method | #API per Image | Require Local Computation | GPU Memory (MB) | Wall-clock Time (s) | Average Accuracy (%) | Gain |
> |--------|----------------|----------------------------|-----------------|---------------------|------------------|------|
> | Source | 1 | - | - | 0.045 | 55.5 |  |
> | LAME | 1 | $\checkmark$ | 2 | 0.046 | 54.1 | -1.4 |
> | ZOO-SPSA | 16 | $\checkmark$ | 52 | 0.450 | 55.1 | -0.4 |
> | Test-time Augmentation | 64 | $\checkmark$ | - | 1.800 | 55.6 | +0.1 |
> | BETA (with ViT-Tiny) | 1 | $\checkmark$ | 1,292 | 0.047 | 58.2 | +2.7 |
> | BETA (with ViT-Small) | 1 | $\checkmark$ | 2,616 | 0.048 | 62.6 | +7.1 |
>
>
>
> This analysis reveals two key points:
>
> 1. **Local Compute is Negligible**. As the table shows, all methods (except Source) require some local computation. Our BETA framework (with ViT-Small) requires 2.6GB of GPU memory, a trivial amount that can be handled by any modern GPU, including consumer-grade cards. Furthermore, the local time overhead is also negligible. The true bottleneck is the API call itself (0.045s). Our method's local computation (forward pass parallelized with API wait + backpropagation) adds only 0.003s per image (0.048s for BETA vs. 0.045s for Source).
>
> 2. **API Calls Dominate Latency:**. The primary factor in wall-clock time is the **number of API calls**. ZOO (16 calls) and Test-time Augmentation (64 calls) are **9.4x** (0.450s / 0.048s) and **37.5x** (1.800s / 0.048s) slower than BETA per image. In a real-world scenario, the true bottleneck is **network latency**, which is determined by the number of sequential network round-trips. Commercial APIs often have a maximum batch size (e.g., 128 for Clarifai). To process a single batch of 64 test images:
>     - **BETA (Ours) & LAME**: Require 64 total API calls (1 per image). This fits perfectly into 1 batch request.
>     - **ZOO**: Requires $64 \times 16 = 1024$ total API calls. This would require $1024 / 128 =$ 8 sequential batch requests.
>     - **Test-time Augmentation**: Requires $64 \times 64 = 4096$ total API calls. This would require $4096 / 128 =$ 32 sequential batch requests.
> 3. **Context for "Backpropagation-Free" Methods**. This analysis also clarifies the context of "backpropagation-free" approaches. In typical white-box TTA, skipping backpropagation is beneficial because the backward pass on a large local model is slow. In our black-box setting, **the primary bottleneck is the API forward pass** (0.045s), which is dominated by network latency. The local backward pass, which only runs on our tiny steering model, is comparatively instantaneous (0.003s). Therefore, eliminating this negligible 0.003s step would provide no practical speed benefit, as the total time is dictated by the API call.
>
> In conclusion, this analysis confirms BETA is highly efficient. It achieves a +7.1% performance gain while being 16x-64x more query-efficient than other input-adaptation/augmentation methods and adding only a negligible (0.003s) local computational overhead per image.

---

> ### Author Response · Authors · 2025-11-21
> **Response to Reviewer 2jik [4/4]**
>
> ## **W5 & Q4: Stability/Robustness Analysis**
> ---
> While our paper originally focused on "optimization stability" (avoiding the collapse seen in ZOO), we agree that analyzing robustness to testing conditions like batch size and data distribution is equally critical.
>
> Following the reviewer's advice, we extended our evaluation to include: (1) **Varying Batch Sizes**, (2) **Label Imbalance** (following [f]), and (3) **Continual Domain Shifts** (following the protocols in [1]).
>
> 1. **Robustness to Batch Size**. We evaluated BETA with batch sizes ranging from 4 to 128. As shown in the table below, BETA is highly robust. Even with a very small batch size of 4, it achieves 59.3%, significantly improving the Source model (55.5%).
>
> | Batch Size | Source | 4 | 8 | 16 | 32 | 64 | 128 |
> |--------|----|----|----|-----|-----|-----|------|
> | Average Accuracy | 55.5|59.3 | 60.1 | 62.3 | 62.5 | 62.6 | 62.6 |
>
> 2. **Robustness to Label Imbalance and Continual Shifts** We further evaluated BETA under (1). **Label Imbalance**: The class distribution within batches is highly skewed (following [f]); (2)**Continual Changing Domains**: The model adapts to a stream of different corruption types sequentially without resetting  (following [1]). As the below results show, BETA maintains strong performance in these challenging scenarios (61.8% and 61.5%), with only a minor drop compared to the standard i.i.d. setting (62.6%).
>
> |Method|gauss.|shot|impul.|defoc.|glass|motion|zoom|snow|frost|fog|bright.|contrast|elastic|pixelate|jpeg|Avg Acc|
> |-|-|-|-|-|-|-|-|-|-|-|-|-|--|-|-|-|
> | Source | 56.8 | 56.8 | 57.5 | 46.9 | 35.6 | 53.1 | 44.8 | 62.2 | 62.5 | 65.7 | 77.7 | 32.6 | 46.0 | 67.0 | 67.6 | 55.5 |
> | BETA | 60.5 | 60.7 | 61.1 | 54.5 | 52.2 | 59.9 | 56.3 | 63.6 | 64.7 | 66.1 | 78.1 | 53.4 | 62.1 | 73.3 | 72.0 | 62.6 |
> | BETA (**in Label Imbalance Setting [f]**) | 59.0 | 59.9 | 59.5 | 53.9 | 51.1 | 59.1 | 55.5 | 62.9 | 64.3 | 65.4 | 77.9 | 52.4 | 61.2 | 73.1 | 72.1 | 61.8 |
> | BETA (**in Continual Shifts Setting [1]**) | 59.5 | 61.0 | 60.4 | 52.3 | 51.4 | 58.4 | 55.2 | 61.8 | 63.3 | 63.8 | 77.4 | 51.8 | 61.7 | 72.5 | 71.3 | 61.5 |
>
>
> 3. **Why is BETA robust**? This robustness is intuitive given our design. Unlike white-box TTA methods that update the model's internal parameters (which risk catastrophic forgetting or overfitting to biased batches), BETA keeps the black-box model's weights frozen. We only learn an additive input prompt. Furthermore, the local steering model is updated with a small learning rate and strong regularization, preventing it from overfitting to dynamic changes in the data stream. This makes BETA naturally resilient to the instability issues often seen in dynamic test-time adaptation.
>
> [1] Continual Test-Time Domain Adaptation, CVPR 2022.
>
> ## **W6 & Q5: Steering Model Initialization & Pre-training Assumptions**
> ---
> We clarify that our method relies only on **standard public initialization** and does not require the steering model's pre-training data to match the target model's.
> 1. **Setup**. In our strictly defined black-box setting, we assume the perspective of **a normal user** accessing a public API. The user typically knows the model's functionality (e.g., classification) but has no access to the provider's private pre-training data (e.g., the massive proprietary datasets used by Clarifai or OpenAI). Consistent with this realistic setting, in all our main experiments, the local steering model (e.g., ViT-Small/16) is initialized with **standard weights pre-trained on ImageNet-1K**. We do not perform any special pre-training or domain alignment to match the black-box model before the test-time adaptation process begins.
> 2. **Evidence**. Regarding the reviewer's query on whether datasets must match, our experiments provide strong evidence that **they do not need to match**.
>     - **Evidence from CLIP Experiments**: As shown in **Table 4**, we successfully use the standard ImageNet-1K trained ViT-Small steering model to adapt a black-box CLIP-B/16 model. The black-box CLIP model was trained on OpenAI's massive, private dataset of **400 million image-text pairs**, which differs fundamentally in size, distribution, and supervision type from the steering model's data.
>     - **Implication**: Despite this massive discrepancy, BETA successfully improves the CLIP model's performance (e.g., raising average accuracy on ImageNet-S/R to 63.4%).
>
> This demonstrates that the steering model acts as a provider of general visual feature gradients. As long as the steering model has learned reasonable visual representations (which are readily available in standard public checkpoints), it can effectively guide the adaptation of a black-box model trained on completely different, private data. Thus, our framework does **not** assume the user has access to the black-box model's pre-training data.

---

> ### Author Response · Authors · 2025-11-27
> **Follow-up on rebuttal**
>
> Dear Reviewer 2jik,
>
> Thanks again for your time and effort. As the discussion period is ending soon, we wanted to ensure you saw the new experiments we ran based on your suggestions.
>
> Specifically, we added **augmentation-based baselines (ZERO)** and expanded our evaluation to the **full set of ImageNet variants** and **EuroSAT**. Results confirm BETA outperforms ZERO while being more API efficient (1 vs. 448 calls). We also validated BETA's **computational efficiency** and **robustness to batch size and ordering**.
>
> We hope these results address your concerns regarding comparisons, efficiency, and robustness. Please let us know if you have any remaining questions; we are happy to provide further clarification.

---

### Official Review · Reviewer_LZF6 · 2025-11-11

**Soundness:** 2
**Presentation:** 3
**Contribution:** 2
**Rating:** 2
**Confidence:** 3

**Summary:**

This paper introduces a novel framework, BETA, designed to address the challenge of adapting pre-trained models in the black-box setting, where only API access is available. The algorithm itself is quite innovative, particularly in how it combines prediction harmonization, steering model regularization, and visual prompt optimization to achieve efficient adaptation. The design of the framework is well thought out, and it successfully balances the trade-offs between optimizing the visual prompt and ensuring model stability during test-time adaptation.

**Strengths:**

(1) Novel Algorithm Design: The combination of prediction harmonization, steering model fine-tuning, and entropy-based filtering introduces an innovative way to improve black-box adaptation. This approach is technically sound and presents a significant improvement over existing methods like ZOO, LAME, and other black-box TTA strategies.

(2) Stabilization Mechanisms: The use of consistency regularization and non-redundant sample filtering addresses the common issue of instability in adaptation, ensuring more reliable performance.

**Weaknesses:**

However, I have serious concerns regarding the paper's claim that it solves the black-box adaptation problem. The authors state:
"In this work, we addressed the critical challenge of adapting powerful models in the strict black-box setting where only API access is available."

While the approach is indeed innovative, there is a major inconsistency between this claim and the method proposed. The key issue is that the authors rely on a steering model that is pre-trained on source data, which contradicts the idea of black-box adaptation. In the strict black-box setting, the very essence is to prevent access to model parameters to avoid data leakage, model distillation, and related security risks.

However, the introduction of a steering model—which is an accessible, pre-trained model—is essentially a workaround that compromises the security and privacy principles of black-box models. The steering model, though lightweight, still allows access to certain information that the black-box framework intends to protect. This introduces a critical flaw, as real-world black-box models (such as those deployed in secure environments) are designed specifically to avoid such exposure. Thus, the approach outlined in this paper is unlikely to be directly applicable in many practical scenarios where security and privacy are paramount.

**Questions:**

See weaknesses

---

> ### Author Response · Authors · 2025-11-21
> **Response to Reviewer LZF6**
>
> We sincerely thank Reviewer LZF6 for their valuable feedback and for recognizing the innovative algorithm design of BETA. We are glad the reviewer found our framework, which combines prediction harmonization, prompt optimization, and stabilization mechanisms, to be "**technically sound**" and a "**significant improvement**" over existing methods.
>
> **Below, we address the concern regarding the consistency of our method with the strict black-box setting (also in the revised manuscript lines 84-88).**
>
> ## **W1: Adherence to the Strict Black-Box Setting and Privacy Security**
> ---
> The reviewer expresses concern that using a steering model "pre-trained on source data" might compromise the security and privacy principles of black-box models. We agree entirely that strict black-box adaptation must prevent data leakage and model distillation.
>
> **We believe this concern arises from a need to clarify the *location* and *nature* of the steering model.** BETA operates **fully** within strict black-box constraints and introduces zero security risks to the target model.
>
> ### 1. The Security Boundary: Client-Side Tool vs. Server-Side API
> ---
> The core distinction lies in where the models reside:
> - **Target Model (Server-Side)**: This is the proprietary, opaque API (e.g., GPT-4o, Clarifai) located on the provider's server. We treat this as a **locked box**; we have zero access to its parameters, gradients, or architecture.
> - **Steering Model (Client-Side)**: This is a separate, local white-box model (e.g., a standard ViT-S/16) that the user runs on their own machine. **Crucially, there is no connection between the two**. The steering model acts purely as a local "flashlight" to guide the input prompt optimization. It never accesses the internals of the black-box API, nor does it require the API to expose any information beyond standard prediction probabilities.
>
>
> ### 2. Public Data vs. Proprietary Data (Addressing "Source Data" Concerns)
> ---
> The reviewer mentions the risk of using a model "pre-trained on source data." **We emphasize that the steering model relies only on public data, not the black-box's proprietary data**.
> - **Public Knowledge**: The steering model is a standard **off-the-shelf** checkpoint (e.g., from HuggingFace or timm) trained on public datasets like ImageNet.
> - **Proprietary Secrets**: The black-box model is trained on the provider's massive, private datasets.
> - **BETA does not require the steering model to match the black-box's training data**. In fact, **Table 6** demonstrates that we can successfully use a standard vision-only ViT (pre-trained on **ImageNet**) to steer a powerful **CLIP Vision-Language Model** (pre-trained on **OpenAI's private dataset of 400 million pairs**), despite their training data being completely different. This proves the steering model is a **generic, public tool**, not a "backdoor" to the provider's secrets.
>
>
> ### 3. Contrast with "Gray-Box" Risks
> ---
> **BETA avoids the exact security risks present in Gray-Box methods**. As defined in **Table 1**, gray-box methods (e.g., FOA, B$^2$TPT) require access to **internal tokens or intermediate features**. We agree with the reviewer that such access violates security protocols. **In contrast, BETA requires strictly zero internal access**, adhering to the privacy standards of commercial APIs.
>
> ### 4. Empirical Proof: Real-World Secure API
> ---
> To validate this security compliance, we deployed BETA on the real-world commercial Clarifai API.
> - In this real-world scenario, we had absolutely **no access** to the model’s parameters or gradients.
> - BETA achieved a **+5.2% accuracy gain** with a budget of just **$0.40** (approx. 120 queries), achieving a **250x cost advantage** over ZOO methods.
>
> **This experiment serves as empirical proof that BETA is directly applicable to secure, proprietary environments without violating any privacy or access protocols.**

---

> ### Author Response · Authors · 2025-11-27
> **Follow-up on rebuttal**
>
> Dear Reviewer LZF6,
>
> Thank you again for your time and effort in reviewing our paper. As the discussion period is ending soon, we wanted to ensure our response successfully clarified your security and privacy concerns.
>
> We explained that BETA runs entirely **client-side** using **public data**, ensuring **zero risk to proprietary server parameters**, a fact we **validated on a real-world commercial API**.
>
> We hope this resolves your concerns about the validity of the black-box setting. Please let us know if you have any remaining questions; we are happy to provide further clarification.

---

### Author Response · Authors · 2025-12-03
**Rebuttal Summary: Reviewer Consensus and New Evidence [2/2]**

### **5. Misunderstanding: Problem Setting & Accessibility (Reviewers LZF6, BkkH)**

* **Misunderstanding:** Reviewer LZF6 questioned the validity of the black-box setting (privacy concerns), and BkkH requested clarification on "Logit vs. Probability" access.
* **Our Response:** We clarified that we assume the role of a **normal user** accessing a public API. We send raw input and receive the **Softmax Probability Vector** (standard for real-world API provided by OpenAI/Clarifai), *not* Raw Logits. The steering model resides strictly **Client-Side** and never accesses server-side parameters and data.
* **Reviewer Consensus:** While LZF6 expressed concern, other reviewers strongly validated this setting. Reviewer 2jik praised the **"clear formalization of the problem"**, Reviewer RoQK called the setting **"unique" and "very useful"**, and Reviewer BkkH highlighted it as an **"important, promising, yet under-explored problem"**. We believe this consensus outweighs the isolated concern.

### **6. Misunderstanding: Contribution & Offline vs. Online (Reviewer BkkH)**

* **Misunderstanding:** The reviewer initially conflated our "Unsupervised Online TTA" setting with "Supervised Offline Few-Shot Transfer" (e.g., BlackVIP), leading to incorrect cost comparisons.
* **Our Response:** We clarified that BETA operates in a **Source-Free Unsupervised Online** setting (unlabeled stream, 0-shot), whereas offline methods require labeled support sets and ~8 million API calls for training.
* **Reviewer Confirmation:** Reviewer BkkH accepted this distinction and explicitly recognized the value of our contribution, stating: *"I have just reminded myself that your main contribution is designing a novel unsupervised learning objective!! ... Some of my concerns are addressed, and I want to raise my score accordingly."*

### **7. Clarification: Steering Model Initialization (Reviewers LZF6, 2jik)**

* **Concern:** Reviewers asked if the steering model requires private source data, which would violate privacy constraints.
* **Our Response:** We clarified that the steering model is initialized strictly from **Public Data** (e.g., ImageNet). We empirically proved this by successfully adapting a black-box CLIP (trained on ~400 million private proprietary data) using a standard public ViT-Small and further validating it on the real-world Clarifai API.

### **8. Conclusion**
Overall, our paper aims to address the inefficiency and instability of black-box Test-Time Adaptation, providing a practical solution for adapting powerful APIs without access to gradients or internal parameters with unlabeled test data. Our main novelty lies in the **BETA** framework, which leverages a lightweight, local white-box steering model to enable a tractable optimization pathway via **prediction harmonization** (the unsupervised learning objective), stabilized by **consistency regularization** and **prompt-oriented filtering**. Extensive empirical results on both standard Vision Models and Vision-Language Models, including a real-world commercial API validation demonstrating a **250x cost advantage**, confirm that BETA achieves state-of-the-art black-box performance with **single-query efficiency**.

We deeply appreciate the reviewers' constructive engagement, which has significantly strengthened this work, as reflected in the updated manuscript. We are encouraged that the successful resolution of technical concerns via new experimental evidence led to the explicit confirmation of score raises from Reviewers RoQK and BkkH. We respectfully submit this consensus for your consideration as you formulate your final recommendation.

Finally, we reiterate our sincere appreciation to all Reviewers, ACs, SACs, and PCs for their diligence and dedication throughout this challenging review cycle.

Best regards,

The Authors

---

### Author Response · Authors · 2025-12-03
**Rebuttal Summary: Reviewer Consensus and New Evidence [1/2]**

Dear Reviewers, ACs, SACs, and PCs,

We sincerely thank you for your dedication and effort in managing this review process under the current challenging circumstances, and we express our deep gratitude to the reviewers for their constructive feedback that has significantly strengthened our work. We understand the additional workload required to assess this submission following the system reset. To assist your review, we summarize the status of the discussion. Prior to the reset, **two reviewers (RoQK, BkkH) explicitly confirmed their intent to raise scores**. These decisions were driven by specific technical resolutions and extensive new experiments detailed below.

### **1. Concern: Missing Stronger Baselines (Reviewers RoQK, 2jik)**

* **Weakness:** Reviewers requested comparisons against stronger baselines, specifically Diffusion-based methods (DDA) and Augmentation-based methods (ZERO).
* **Our Response:** We implemented both **DDA** and **ZERO**. The results demonstrate that BETA outperforms both:
  * **vs. DDA (Diffusion):** BETA is **200x faster** (0.05s vs 10.8s/img) and achieves higher accuracy (+7.1% vs +1.4%).
  * **vs. ZERO (Augmentation):** BETA achieves higher accuracy (66.5% vs 63.7%) with far greater efficiency (1 API call vs 64 calls).
* **Reviewer Confirmation:** Reviewer RoQK validated these results, stating: *"Given that my concerns have been addressed, I raise my score"* to **8**.

### **2. Concern: Efficiency of Both API and Local Compute (Reviewers RoQK, 2jik, BkkH)**

* **Weakness:** Reviewers questioned the trade-off between API costs and local compute, and whether the method is feasible for real-time applications.
* **Our Response:** We provide a detailed breakdown demonstrating that on the real-world black-box Clarifai API, BETA achieves superior performance while being **16x–200x more query-efficient** and **~10x–40x faster** in wall-clock time than baselines. We also clarify that the steering model is lightweight, requiring only 2.6GB of GPU memory (feasible for consumer GPUs) with negligible local latency (0.003s). Furthermore, when evaluated under strict real-time constraints where slow methods are forced to skip data, BETA maintains 62.5% accuracy (a negligible 0.1% drop), whereas ZOO collapses to 54.3%—performing worse than Source due to adaptation lag.
* **Reviewer Confirmation:** Reviewer BkkH accepted the cost clarification. Reviewer RoQK accepted the efficiency and real-time analysis, explicitly raising their score.

### **3. Concern: Robustness in Challenging Settings (Reviewers RoQK, 2jik)**

* **Weakness:** Reviewers requested evaluation under more challenging conditions, including varying batch sizes, label imbalance, and continual domain shifts.
* **Our Response:** We extended evaluation to include **Label Imbalance** and **Continual Domain Shifts**, where BETA maintained high accuracy (61.8%) with minimal degradation. We also demonstrated robustness to small **Batch Sizes** (stable down to batch size 4).
* **Reviewer Confirmation:** Reviewer RoQK found these results convincing and explicitly raised their score.

### **4. Concern: Evaluation on Additional Datasets (Reviewer 2jik)**

* **Weakness:** Reviewer 2jik specifically requested evaluation on a broader set of benchmarks, including the "full set of ImageNet variants" and "fine-grained datasets".
* **Our Response:** We expanded our evaluation to include the full **ImageNet suite (S/R/A/v2)** and **EuroSAT** with CLIP. The results were compelling: on the fine-grained EuroSAT dataset, BETA achieved a **+11.3% gain** with a single API call, significantly outperforming query-intensive baselines.
* **Result:** These additional results directly addressed Reviewer 2jik’s request.

---

### Meta-Review · Area_Chair_yW14 · 2025-12-24

**Summary:**

This paper was reviewed by four experts in the field, and the reviews are mixed. The paper received a negative initial score ranging from Reject (2, 2, 2) to Marginal Accept (6). While several reviewers appreciate the importance of the problem setting and find the idea promising, there is not sufficient consensus in favor of acceptance.

This article presents BETA, a new technical framework for adapting pre-trained models in a "black box" context, where only API access is available. The models in this framework are then considered an API. The article formalizes this framework, which is new to my knowledge, and proposes a model, BETA, that addresses this problem. BETA instantiates a local  local white-box steering model  to create a tractable gradient pathway for optimization to minimize the entropy of the steering model and the original black box model combined.

Therefore, the decision is not to recommend acceptance. The authors are encouraged to carefully consider the reviewers’ feedback.

**Reviewer Concerns:**

The reviewers initially raised the following main concerns:
1. Mismatch with the claimed black-box setting:
A central concern raised by multiple reviewers is that the paper claims to address strict black-box test-time adaptation, yet relies on a pre-trained steering model trained on source data. Reviewers argue that this is problematic for the core black-box assumption, as such a steering model effectively introduces additional accessible information that would not be available in realistic black-box or API-only deployment scenarios, particularly those motivated by security and privacy constraints.
2. Unclear conceptual justification and logical gaps:
Several reviewers highlight weak or insufficient justification for key design choices. The empirical evidence provided (e.g., Figures 2 and 3) is also seen as not convincingly supporting the mechanisms.
3. Method complexity and presentation issues:
The method involves multiple interacting components (steering model, prompt optimization, normalization updates, filtering, consistency losses), which reviewers found difficult to follow. I also found it hard to follow when I read the paper. I really wonder if the paper took the time to correct the paper based on the review. I cannot imagine that the version I was reading to be the version after the correction.
4. Limited and potentially weak baseline comparisons:
The experimental evaluation is considered incomplete by most reviewers. Reviewers point out missing comparisons to relevant black-box TTA and augmentation-based methods.
5. Questionable efficiency and cost claims:
While the paper emphasizes efficiency in terms of API calls, reviewers note that computational cost, memory usage, optimization time, and robustness to testing conditions are not sufficiently analyzed.
6. Limited scope and generality of experiments:
The evaluation focuses on a narrow set of architectures, datasets, and settings. Reviewers suggest that broader evaluation is needed to support the generality of the claims.

**Reviewer Scores:**

After bad initial ratings (three scores of 2 and one of 6), a reviewer raised their score to 8 following the OpenReview publication. Despite the authors' considerable efforts to persuade the reviewers, most of the reviewers did not change their opinion. I admit that, despite this all the rebuttal comments, I still struggle to fully grasp the contribution and the relevance of the issue. I therefore choose to reject the article and advise the authors to take the reviewers' comments into account.

---

### Decision · Program_Chairs · 2026-01-26

Reject